# Ribonuclease 4 Functions in Nociceptor-Mediated Nerve Homeostasis

Xiaona Feng [1,6,8], Kaiwen Zhang [1,8], Prach Techameena [1,8], Rolen M. Quadros[2], Csaba Adori[3,4], Aliia Murtazina[5], Igor Adameyko [5], Sofia Biagini [1], Ozun Gokce Bayramlik [1], Francois Lallemend [4], Channabasavaiah B. Gurumurthy[2,7] & Saida Hadjab [1] ✉

The regulation of nociceptor identity and function is essential, as disruptions can significantly influence pain sensation, yet our understanding of the molecular mechanisms involved remains incomplete. In this study, we identified ribonuclease 4 (RNase4) as selectively expressed in the unmyelinated nociceptor lineage. Analysis of *RNase4*-deficient mice and single-cell transcriptomic data revealed a cell-autonomous role for RNase4 in regulating nociceptor function. Moreover, in a neuropathic pain model, *RNase4* expression was upregulated in nociceptors during the pain and recovery phases, and its deletion altered mechanical sensation. Additionally, RNase4 exerted non-cell-autonomous effects on the myelin structural organization of adjacent myelinated axons. Together, these findings implicate RNase4 as a dual regulator of nociceptor biology and myelin integrity, revealing a molecular pathway for pain regulation and nerve repair.

Nociceptors, the specialized pain-sensing neurons of the peripheral nervous system, are central to the detection and transmission of noxious stimuli to the central nervous system[1]. These neurons, located in the dorsal root and trigeminal ganglia, are equipped with a variety of receptors that detect thermal, mechanical, and chemical noxious stimuli[2], highlighting the molecular complexity of pain perception and the need for cell-type-specific biomarkers that regulate nociceptor identity and function[3,4].

Nociceptors are classified into two primary fiber types: Aδ fibers, which transmit fast, sharp pain, and C fibers, which convey slower, diffuse pain signals[5]. C fibers are further subdivided into distinct subpopulations characterized by the expression of specific molecular markers, including neuropeptides such as CGRP and substance P, receptors like TrkA, P2X3 purinergic receptors, Ret, and Mrg family receptors, as well as ion channels including TRPV1, TRPM8, TRPA1, and ASICs. These molecular signatures form critical cell-to-cell communication pathways that underpin the specialized functions of each nociceptor subtype, facilitating the detection and transmission of specific pain modalities. As a result, their modulation, or genetic deletion, in vivo has been shown to alter pain sensation[1,6]. The mechanisms regulating nociceptor function and their contribution to the progression of neuropathic pain, particularly following nerve injury, remain, however, less understood. Emerging evidence highlights the role of specific transcriptional regulators in these processes. For example, ATF3[7] and SOX11[8] have been implicated in axonal regeneration and pain sensitization, respectively, after peripheral nerve injury. Additionally, the transcriptional repressor G9a[9], which is upregulated following nerve injury, has been associated with nociceptor plasticity underlying neuropathic pain.

[1]Laboratory of Neurobiology of Pain & Therapeutics, Department of Neuroscience, Karolinska Institutet, Stockholm, Sweden. [2]Mouse Genome Engineering Core Facility, University of Nebraska Medical Center, Omaha, NE, USA. [3]Department of Molecular Biosciences - The Wenner-Gren Institute (MBW), Stockholm University, Stockholm, Sweden. [4]Department of Neuroscience, Karolinska Institutet, Stockholm, Sweden. [5]Department of Physiology & Pharmacology, Karolinska Institutet, Stockholm, Sweden. [6]Present address: Institute of Aging, Key Laboratory of Alzheimer's Disease of Zhejiang Province, School of Mental Health and Affiliated Kangning Hospital, The First Affiliated Hospital, Wenzhou Medical University, Wenzhou, Zhejiang, China. [7]Present address: Genome-editing Research, Education, and Animal genome Targeting (GREAT) Core, Department of Cell and Molecular Biology, School of Medicine, University of Mississippi Medical Center, Jackson, MS, USA. [8]These authors contributed equally: Xiaona Feng, Kaiwen Zhang, Prach Techameena. ✉e-mail: saida.hadjab@ki.se

There is still, however, a significant gap in understanding the molecular mechanisms that preserve the homeostatic state of nociceptors and how these are disrupted or contribute to neuropathic pain. This suggests that key regulatory pathways and their roles in nociceptor function and dysfunction remain to be elucidated. To address this, we adopted a broad approach and searched for molecular markers specific to unmyelinated neurons in the DRG, trigeminal ganglia (TG), and cochlea, and identified ribonuclease 4 (RNase4) as a highly selective candidate.

Traditionally, ribonucleases are implicated in RNA metabolism[10], yet they are also involved in a diverse array of physiological processes, including growth, neurogenesis, angiogenesis, immune modulation, and host defense[10–12]. Specifically, it has been shown that RNase4 can be secreted into the extracellular matrix[12]. The restricted expression of RNase4 in unmyelinated nociceptors highlights its potential role in regulating pain pathways and contributing to the functional specialization of these sensory neurons.

Here, we identify RNase4 as a highly selective marker of unmyelinated nociceptors across mouse and human sensory ganglia and uncover an unexpected role for RNase4 in linking nociceptor identity to peripheral nerve homeostasis. By combining single-cell transcriptomics, genetic loss-of-function approaches, behavioral analyses, and nerve injury models, we show that RNase4 acts cell-autonomously to regulate nociceptor transcriptional programs, including potassium channels and pain sensitivity, while exerting non-cell-autonomous control over axonal regeneration, myelin remodeling, and immune-associated repair processes. In vitro analyses further indicate that RNase4 can modulate PI3K–AKT signaling, suggesting a mechanistic pathway through which RNase4 influences these effects. Together, these findings demonstrate that RNase4-expressing nociceptors are not solely pain sensors but function as homeostatic and injury-responsive regulatory hubs that shape peripheral nerve stability, repair, and remodeling, thereby extending the role of nociceptors beyond sensory transduction.

## Results

### Rnase4 is selectively expressed in nociceptors

The identification of genes selectively expressed in nociceptors is crucial for understanding the regulation of pain perception and for developing targeted pain therapies. Using the iPain single-cell transcriptomic atlas[13] to differentiate between various populations of nociceptors and mechanoreceptors[14,15], including peptidergic (PEP), non-peptidergic (NP), somatostatin-positive (SST), and C-low threshold mechanoreceptors (c-LTMRs), in contrast to heavily myelinated fibers (NF, neurofilament positive fibers) (Fig. 1a), we fitted the negative binomial distribution with pyDESeq2[16] on the pseudo-bulk expression and identified differentially expressed genes (DEGs) amongst neuronal subtypes. The top selective genes for nociceptors included *Grik1*, *Syt9*, and *Trpc3*, while *Slc16a6*, *Scn8a*, and *Kank4* were the leading genes for mechanoreceptors, aligning with prior findings[17] (Fig. 1b, Supplementary Data 1). Interestingly, *RNase4* was identified as a selective marker for nociceptor populations (Fig. 1b, c) and was absent from neurons in the CNS[17]. To further characterize this cell type–restricted expression, we performed RNAscope on cross-sections of dorsal root ganglia (DRG) from adult animals (Fig. 1d, j). Co-labeling used established markers: *Calca* (peptidergic nociceptors, PEPs), *Sst* (somatostatin-positive neurons), TH (cLTMRs), NF200/NF-H and PV (myelinated neurofilament neurons, NFs), peripherin (Prph; small-diameter, unmyelinated neurons), and *Prdm12* (nociceptor lineage marker). Quantification of RNase4+ DRG neuron soma size revealed high selectivity in small diameter neurons, with ~80% less than 20 microns and the remaining ~20% being between 20 and 35 microns (Fig. 1e).

RNase4 showed complete overlap with Sst + neurons and partial colocalization (~40–50%) with Calca+ peptidergic neurons (Fig. 1f, g),

while TH+ cLTMRs, which share a developmental nociceptive lineage but mediate pleasant touch, were entirely RNase4-negative (Fig. 1i). Minimal overlap occurred with NF200+ populations and none with PV + myelinated neurons, with RNase4 enriched in Prdm12+/peripherin+ small-diameter neurons (Fig. 1d, h, j). These findings establish RNase4 as a selective marker of specific unmyelinated nociceptor subtypes.

Bulk RNA-seq analysis of human dorsal root ganglia (DRG)[18] confirmed RNase4 expression in human tissue, with a selective nociceptor-enriched pattern evident in a recent single-cell atlas of human somatosensory neurons[19] (Fig. 1k, l). These findings demonstrate evolutionary conservation of RNase4 expression across species, suggesting an important role in nociceptor biology.

### Role of RNase4 on nociceptor phenotype and pain sensation

To investigate how RNase4 influences the identity and functionality of nociceptors, particularly in its absence, we developed a conditional *Rnase4* knock out mouse line (*Rnase4*$^{fl/fl}$) by inserting Short Conditional intrON (SCON) cassette[20] containing LoxP recombination sites, within the *Rnase4* exon and using the Easi-CRISPR technology, a streamlined method utilizing the CRISPR-Cas system for genome editing[21] (Fig. 2a). The *Rnase4*$^{fl/fl}$ mouse line was crossed with the *Baf53b*$^{Cre}$ mouse line to delete *Rnase4* in sensory neurons during embryonic development, after the neurogenesis of nociceptors (Fig. 2b, c). To explore the role of RNase4 on the transcriptional phenotype of nociceptors, we performed RNA sequencing using Smart-seq3xpress on single nuclei from DRG neurons extracted from adult *Rnase4* conditional knockout (cKO) mice (*Baf53b*$^{Cre}$;*RNase4*$^{fl/fl}$, or cKO-BR) and control littermates. After filtering, we obtained 586 nuclei from the cKO-BR line and 514 nuclei from controls, totaling 1,113 neuronal nuclei. We applied cell type annotation using the pre-trained scANVI[22] model from iPain[13] to perform label transfer (Fig. 2d). Approximately 73% of the profiled nuclei were identified as belonging to the nociceptive lineage. A total of 765 genes were found to be significantly differentially expressed (either up- or down-regulated) in the cKO-BR condition compared to controls within the nociceptive lineage neurons (Fig. 2e, f, Supplementary Data 2). Further analysis of the differentially expressed genes (DEGs) using the PANTHER database indicated substantial changes in key pathways and protein classes. Notably, C2H2 zinc finger proteins emerged as the most prominently altered group, with 42 proteins represented. This included various transcription factors such as *Gm6710*, *Zfp433*, and *Maz*. Additionally, increased expression was observed for *Irf3* (interferon regulatory factor) and *Rela* (p65, a component of the NF-κB pathway), suggesting significant alterations in transcriptional regulation and immune response pathways in neurons within the conditional mutant.

The analysis also highlighted notable changes in cytoskeleton-related proteins and ribosomal components (Supplementary Fig. 1a, b). Alterations were observed in microtubule-binding motor protein *Dync2h1*, *Mapt* (microtubule-associated protein tau), and tubulin variant *Tubd1*. Furthermore, eleven ribosomal proteins were affected, including *Mrpl4*, *Rpl9*, *Rps10*, *Rps27*, and *Mrps1*. Significant changes were also noted in translation-related factors such as *Eif2s1* (a translation initiation factor), *Gspt1* (a translation termination factor), and *Cpeb3*, a negative regulator of translation with known function on *Trpv1* expression[23]. These findings indicate widespread modifications in cellular architecture and protein synthesis machinery as a response to conditional mutation. Finally, focusing on ion channels within nociceptors, we identified differences in expression levels of potassium voltage-gated ion channels *Kcnf1* (Kv5.1) and *Kcng3* (Kv6.3) (Fig. 2e, f). These channels are integral to the potassium channel machinery that regulates membrane potential and facilitates the return to a negative resting potential following an action potential. Their reduction in RNase4 cKO neurons may affect Kv channel composition and potentially influence neuronal excitability[24,25]. Interestingly, RNase4 deletion altered gene expression in

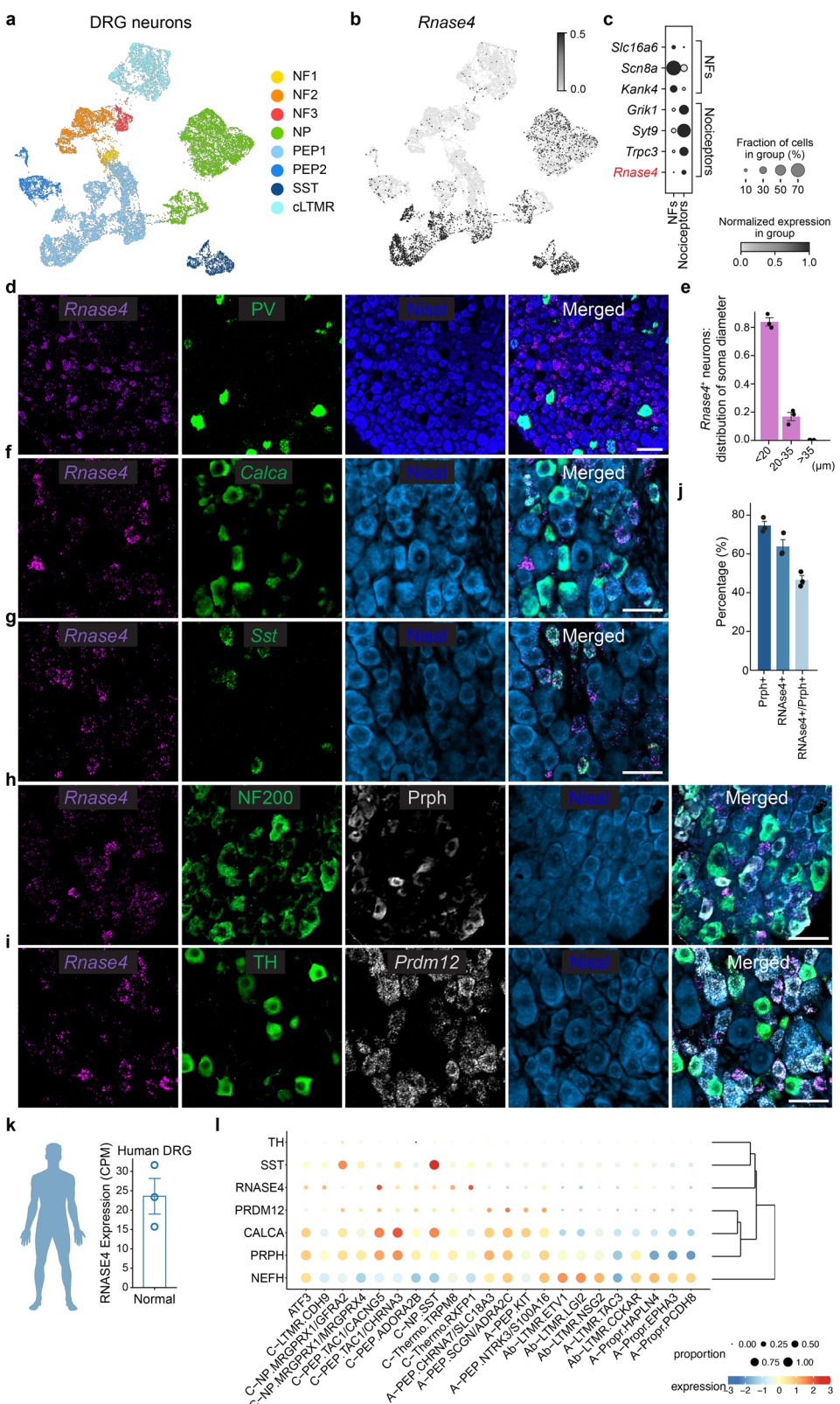

cLTMRs despite the absence of RNase4 in this population, an observation that could be consistent with paracrine effects of RNase4 on neighboring cells.

To determine whether the molecular alterations in cKO-BR mice are associated with changes in behavioral responses to noxious stimuli, we assessed acute responses to thermal and mechanical noxious stimuli. Our results revealed that early neuronal deletion of *Rnase4* during development did not significantly alter mechanical or thermal sensory thresholds compared to control littermates (Fig. 2g). These findings suggest that, despite alterations in gene expression, baseline pain sensation under normal conditions remains unaffected by the absence of RNase4 from early development.

**Fig. 1 | Neuronal Subtypes and RNase4 Expression in the DRG. a** A UMAP plot of neuronal cells in the DRG colored by the respective subtypes. **b** A UMAP plot of neuronal cells in the DRG colored by the expression of *Rnase4*. **c** A dot plot of the top 3 markers distinguishing nociceptors and NFs, including *Rnase4*. **d** A representative DRG section from adult WT mice labelled with *Rnase4* and PV (*Pvalb*) by RNAscope and counterstained with Nissl, scale bar = 50 μm, total *n* = 3 mice. **e** A bar plot representing the mean ± SEM error bar of the proportion of neuronal cell number expressing *Rnase4* according to their respective sizes (*n* = 3 mice).

**f–i** Representative DRG sections from adult WT mice labelled with *Rnase4* and markers for nociceptive subtypes (scale bar =50 μm), total *n* = 3 mice. **j** A bar plot representing the mean ± SEM error for the proportion of cells with positive signal for the canonical markers over Nissl⁺ (quantified from *n* = 3 mice with a minimum of 3 sections per mouse). **k** A bar plot representing the mean ± SEM error bar of *RNASE4* expression in human DRG (*n* = 3 donors). **l** Dot plot representing the expression of canonical neuronal markers in human DRG data from Bhuiyan et al., 2025. Source data are provided as a Source Data file.

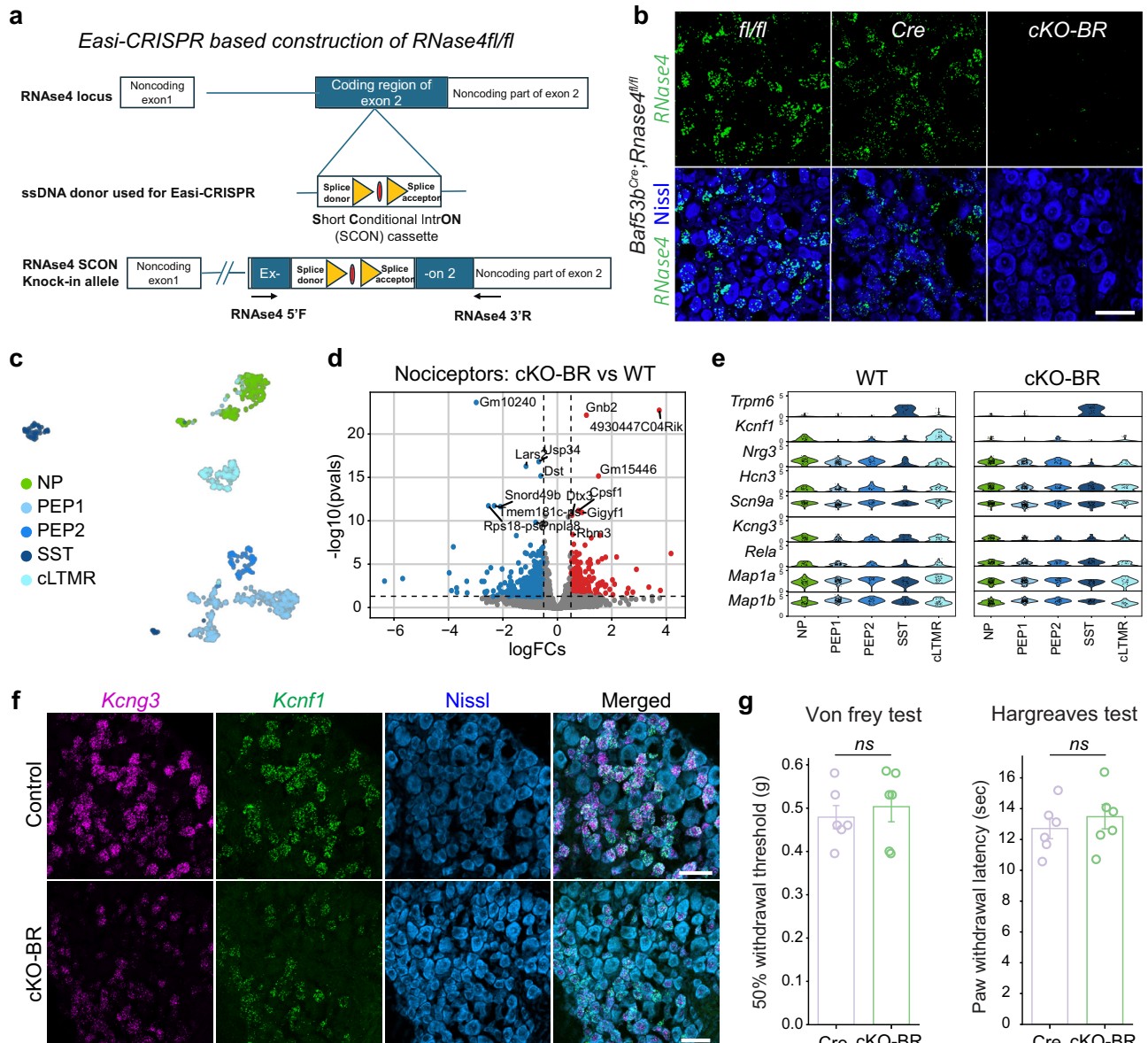

**Fig. 2 | Impact on neuronal molecular changes and pain behavior in RNAse4 conditional knockout. a** Schematic of exon splitting to insert the SCON cassette. RNAse4 locus showing coding and non-coding regions. The Easi-CRISPR method was used for inserting a Short Conditional intron (SCON) cassette into the second exon of RNAse4 to create a conditional knockout allele (please refer to Methods for more details). Short artificial intron cassette containing "a splice donor, a branch point (red oval) flanked with two LoxP sites (yellow triangles) and a splice acceptor". The location of genotyping primers is shown with arrows. **b** Representative RNAscope images of adult DRG sections labeled for Rnase4 (top) and Rnase4 counterstained with Nissl (bottom) in mice of three genotypes: *Rnase4^{fl/fl}* (*fl/fl*, *n* = 4), *Baf53b^{Cre}* (Cre, *n* = 3), and *Baf53b^{Cre}*; *Rnase4^{fl/fl}* (cKO-BR, *n* = 6), scale bar = 50 μm. **c** UMAP plot of neuronal cells in the DRG colored by the respective

subtypes. **d** Volcano plot of differentially expressed genes comparing cKO-BR to WT. The p-values were computed with a two-tailed Wald test, and multiple hypothesis testing was corrected with Benjamini-Hochberg. **e** Violin plots showing the expression of differentially expressed genes (including ion-channels) of WT (left) and cKO-BR (right). **f** Representative DRG sections with scale bar = 50 μm from adult Control (*n* = 3 *fl/fl* and Cre) and cKO-BR (*n* = 3) mice labelled with differentially expressed potassium channels (*Kcng3* and *Kcnf1*). **g** Bar plots of the mean ± SEM error bar representing the mechanical sensitivity (left) and paw withdrawal latency (right) pain-like behavior tests of Cre and cKO-BR. A two-tailed independent t-test was used for the statistical comparisons (6 mice, including 3 male and 3 females, each group was used, ns = no significant). Source data are provided as a Source Data file.

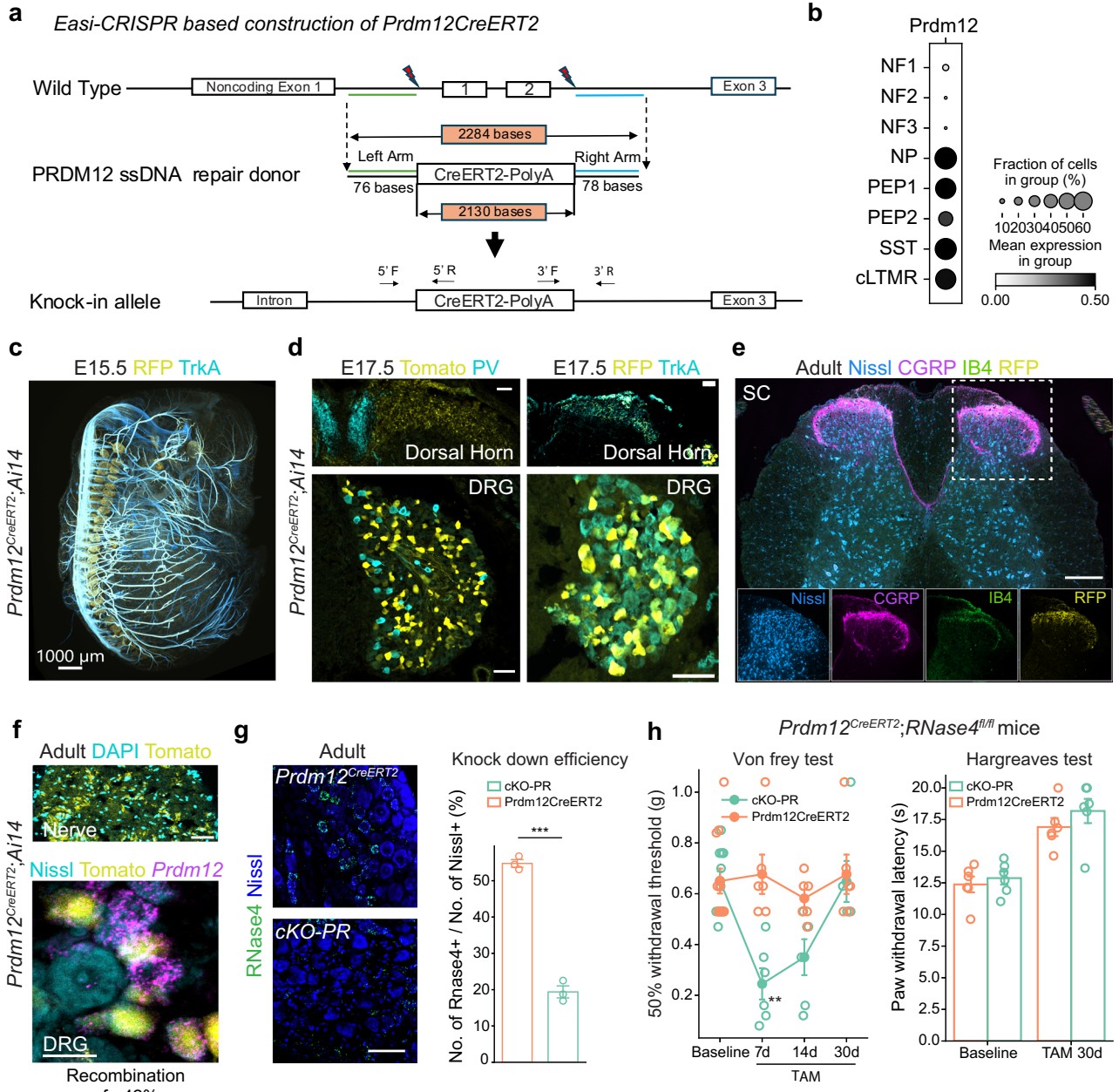

**Fig. 3 | Characterization of *Prdm12^CreERT2* mouse line and *RNase4* recombination in neurons and impact on pain behavior. a** Schematic of *Prdm12^CreERT2* mouse construction. **b** Dot plot of DRG neurons showing *Prdm12* expression in different neuronal subtypes. **c** Whole E15.5 *Prdm12^CreERT2*;Ai14 mouse embryo imaged using the iDISCO protocol for *n* = 1 embryo, stained for tdTomato and TrkA. Merged channels of the whole embryo viewed dorsally (scale bar = 1000 μm). **d** E17.5 *Prdm12^CreERT2*;Ai14 mouse spinal cord (top) and DRG (bottom) stained for PV (left) and TrkA (right), scale bar = 50 μm), representative from *n* = 4 embryos. **e** Adult *Prdm12^CreERT2*; Ai14 mouse spinal cord (top, scale bar = 200 μm) stained with different markers (bottom), representative from *n* = 3 mice. **f** Adult *Prdm12^CreERT2*; Ai14 mouse sciatic nerve cross-section (top, scale bar = 50 μm), and DRG (bottom, scale bar = 20 μm) stained for *Prdm12*, representative from *n* = 4 mice. **g** Left, RNAscope images of adult DRG sections labeled for RNase4 and counterstained with Nissl in *Prdm12^CreERT2* and cKO-PR mice, 7 days post-tamoxifen injection. scale bar = 50 μm. Right, Bar plot of mean ± SEM showing recombination efficiency of the *Prdm12^CreERT2* line (*n* = 3 for *Prdm12^CreERT2* and *n* = 3 for cKO-PR; two-tailed independent t-test was done; ***p < 0.001; p = 5.52e⁻⁵). **h** Left, line plot of mean ± SEM showing mechanical sensitivity in *Prdm12^CreERT2* control and cKO-PR mice at baseline, 7-, 14-, and 30-days post-tamoxifen TAM injection. Right, Bar plot of mean ± SEM representing the paw withdrawal latency of the Hargreaves test from *Prdm12^CreERT2* and cKO-PR mice at baseline and 30 days post-tamoxifen injection. n = 6 mice (3 males and 3 females) in each group. Statistical significance between groups was determined using a two-tailed independent t-test (**p < 0.01; p = 0.0057). Source data are provided as a Source Data file.

## Acute RNase4 deletion in adults transiently increases mechanical pain sensitivity

To investigate the role of RNase4 in adult nociceptors, we employed an intersectional genetic deletion approach. Specifically, we generated a *Prdm12^CreERT2* mouse line (see Fig. 3a and Methods for details), where *PRDM12* serves as a specific marker for the nociceptive lineage[26]

(Fig. 3b, c), to conditionally delete *Rnase4* from nociceptors upon tamoxifen administration (Fig. 3a). The specificity and efficacy of the *Prdm12^CreERT2* line were first validated by crossing it with a *Rosa26^tdTOM* (Ai14) reporter mouse. Following induction, tdTomato fluorescence was detected exclusively in *Prdm12⁺* nociceptors and their projections: RFP+ cells and fibers show clear exclusion with PV (Fig. 3d) and co-

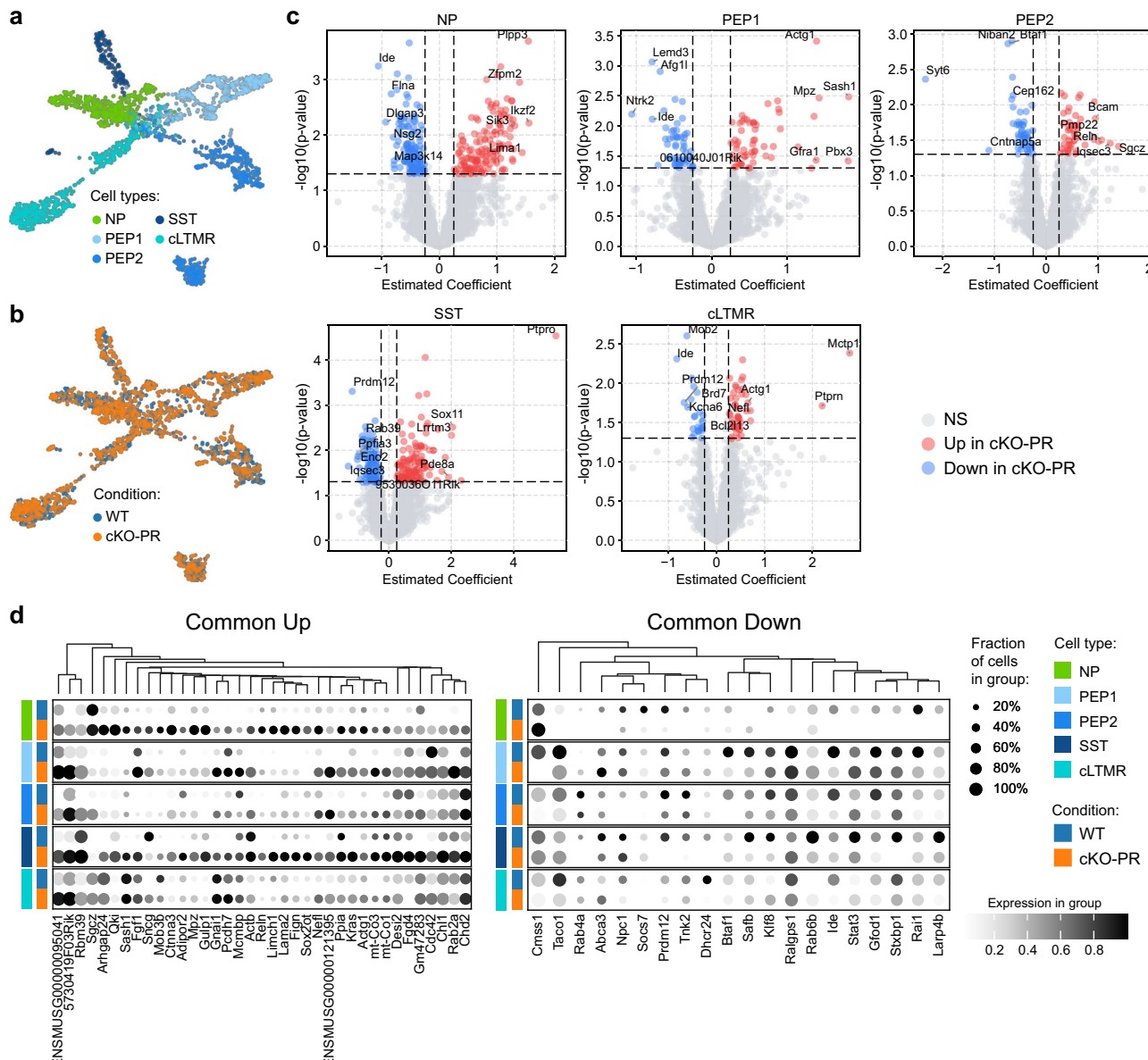

**Fig. 4 | Molecular profiling of *Prdm12*$^{CreERT2}$; *Rnase4*$^{fl/fl}$ Nociceptors.** UMAP plots of nociceptors from the DRG of Prdm12$^{CreERT2}$ and *Prdm12*$^{CreERT2}$; *Rnase4*$^{fl/fl}$ mouse lines, colored by the nociceptive subtypes (**a**) and mouse genotypes (**b**). **c** Volcano plots of differentially expressed genes comparing between cKO-PR and WT for NP, PEP1, PEP2, SST, and cLTMR. The differentially expressed genes were identified with PyDESeq2 where the single-nucleotide data were binned to create pseudo-bulk with 5 pseudo-replicates for each cell type and condition. The p-values were computed with a two-tailed Wald test. Due to the limited sample size resulting from the pseudobulk aggregation, the standard Benjamini-Hochberg multiple hypothesis testing correction was not used. The final gene list was instead generated with consideration of unadjusted p-value < 0.05 and effect size > 0.25. **d** Dot plots of commonly up-regulated (left) and down-regulated (right) genes in the cKO-PR, representing subtypes of nociceptors and mouse genotypes.

localization with TrkA (Fig. 3d), IB4 and CGRP (Fig. 3e), indicating that they terminate within laminae associated with nociceptive afferents. This confirms the selective projection of PRDM12-lineage neurons to I-II regions of the dorsal horn. These findings were consistent across both developmental and adult stages (Fig. 3d, e, Supplementary Fig. 2), with a recombination efficiency estimated at ~42.5% (Fig. 3f). Finally, the successful removal of *Rnase4* was confirmed in adult DRG cross sections from tamoxifen-treated *Prdm12*$^{CreERT2}$; *Rnase4*$^{fl/fl}$ mice (Fig. 3g). We therefore assessed the mechanical withdrawal threshold in our inducible KO (cKO-PR) mice using the von Frey test. Following induction, cKO-PR mice exhibited a significant reduction in the paw withdrawal threshold (Fig. 3h), indicating an increased sensitivity to mechanical stimuli. Importantly, this change in threshold was transient: by day 14 post-tamoxifen injection, cKO-PR mice began to

recover their baseline sensitivity, with full recovery observed by day 30. Thermal sensitivity, assessed with the Hargreaves test, showed no genotype-dependent differences at the intermediate time points; therefore, only the 30-day Hargreaves data are presented. These findings suggest RNase4 contributes to adult nociceptor function, although its absence causes only transient effects (likely due to compensatory mechanisms). The data also suggest a role for RNase4 in nociceptor homeostasis. This concept could be particularly relevant in models of chronic pain, where such homeostasis is disrupted.

To further characterize molecular changes underlying the cKO-PR phenotype, we performed single-nucleus RNA sequencing (snRNA-seq) using the 10x Genomics platform on dorsal root ganglia (DRG) from cKO-PR mice and wild-type controls (Fig. 4a, b). Dimensionality reduction and clustering revealed distinct neuronal subtypes, with

nociceptors showing the most pronounced transcriptional alterations in cKO-PR samples (Fig. 4b). Differential expression analysis identified a core set of significantly downregulated genes shared across subtypes, including the nociceptor-specific transcription factor *Prdm12* (Fig. 4c; Supplementary Data 2). Nociceptor-specific profiling revealed subtype-selective up- and downregulation of genes (Fig. 4d). Among the list of differentially expressed candidates (Supplementary Data 2), we were able to extract several ion channels such as *Kctd1* and *Kcnh8*, that were upregulated in cKO-PR, consistent with potential mechanisms of neuronal hyperexcitability contributing to the behavioral phenotype.

## Non-cell-autonomous action of RNase4: modulation of PI3K−AKT−mTOR signaling

To examine whether extracellular RNase4 (R4) engages intracellular signaling pathways, ND7/23 cells were treated with RNase4 alone or in combination with inhibitors of the PI3K−Akt−mTOR pathway, a central signaling axis linking extracellular cues to neuronal regulation. Pathway activation was assessed by measuring phosphorylation of Akt (Ser473) and mTOR (Ser2448), established readouts of pathway activity[27].

In the first experimental dataset (Fig. 5a), inhibition of Akt (MK-2206) and PI3K (LY294002) markedly reduced p-Akt levels compared with vehicle control, confirming effective pathway suppression. When ND7/23 cells were co-treated with RNase4, p-Akt levels remained similarly reduced in all inhibitor conditions, indicating that RNase4 does not override pharmacological pathway blockade. Notably, RNase4 decreased p-Akt/Akt ratios relative to DMSO alone, suggesting that extracellular RNase4 can dampen basal Akt activation. This reduction corresponds to an estimated 60% (Fig. 5a, b) ($P = 0.0072$; $n = 3$ biological replicates) decrease in the p-Akt/Akt ratio, indicating a significant inhibitory effect of RNase4. Interestingly, while LY294002 robustly and significantly suppressed p-Akt levels, wortmannin led to a decrease of p-Akt level, though it was not significant, and led to a moderate suppression of around 65% (not significant) when associated with RNase4 (Fig. 5b), despite targeting the same pathway. Furthermore, no significant effect of inhibitor rapamycin alone was found at any concentration, though surprisingly, the Akt levels increased by 40% in line with rapamycin-mediated inhibition of mTORC1, which can activate PI3K/Akt signaling by relieving a negative feedback loop[28], which may explain the upward trend in Akt levels observed in the same experiment. The addition of RNase4 to rapamycin-treated cells diminished p-Akt levels to an estimate of 60% compared to rapamycin only ($P = 0.011$ for 5 nM condition and $P = 0.018$ for 10 nM condition, Fig. 5b). These data indicate that RNase4 exerts a consistent Akt-suppressive effect on ND7/23 cells. Investigating the effect of RNase4 on mTOR phosphorylation, particularly at Ser2448 (Fig. 5c, d), we found a subtle decrease of around 20%, though not significant. Expectedly, mTOR level was found to be significantly reduced ($P = 0.0082$) by the PI3K/Akt pathway inhibitor LY294002 (Fig. 5c, d). βIII-tubulin levels were comparable across all conditions (Fig. 5e, f), verifying equal protein loading and supporting the reliability of the phosphorylation analyses.

Notably, while Akt and mTOR expression were detectable, Axl receptor protein, a known PI3K−Akt−mTOR modulator, was undetectable in both ND7/23 cells and primary sensory neuron cultures (Supplementary Fig. 4), indicating that the effects of RNase4 on this pathway in the DRG-derived context are Axl-independent. Taken together, our results demonstrate that RNase4 may also exert its effect through modulating the PI3K−AKT pathway, but its effects here may not be through the Axl receptor.

## Dynamic of Rnase4 expression in a nerve injury model

To examine changes in *Rnase4* expression following nerve injury in a neuropathic pain model, we utilized the iPain database[13]. This resource includes RNA datasets from animals both with and without injury, specifically focusing on those subjected to nerve crush up to four weeks post-injury, as well as control animals representing baseline conditions. Initial mapping of *Rnase4* expression on the *Pain dynamics* (*Reference*, *Pain*, *Recovery* and *Lasting*, Fig. 6a) revealed a significant increase in *Rnase4* levels across all nociceptor subtypes following injury (Fig. 6a–c). Interestingly, the increase in *Rnase4* expression was most prominent during the first week post-injury and was particularly marked in nociceptor populations that initially expressed lower levels of *Rnase4* (NP, PEP2, and cLTMRs). Notably, in these populations, *Rnase4* levels returned to near baseline after two weeks, while in populations that initially expressed higher levels of *Rnase4* (PEP1 and SST), overexpression persisted. We further extended our analysis to human DRG using bulk RNA-seq from[18] and found a similar increase in RNase4 expression in donors diagnosed with pain (Fig. 6d). This trend was, however, not statistically significant, likely due to the limited number of samples.

To predict the interaction network of *Rnase4* during *Pain dynamics*, we applied Spearman's rank correlation analysis to identify genes whose expression was correlated with *Rnase4*. Gene Ontology (GO) analysis of the resulting network revealed distinct pathways associated with RNase4. Under naïve conditions, the top three GO terms for positively correlated genes were "Regulation of Signal Transduction by P53 Class Mediator", "Epithelial Cell Migration", and "Positive Regulation of Transcription by RNA Polymerase II". In contrast, negatively correlated genes were associated with "Chemical Synaptic Transmission", "Cell Junction Assembly", and "Cell-Cell Adhesion Mediated by Cadherin" (Supplementary Fig. 5a, Supplementary Data 3). Fourteen days after nerve crush injury, the GO terms for positively correlated genes shifted to "Cellular Response to Organic Cyclic Compound", "Response to Axon Injury", and "Nervous System Development". For negatively correlated genes, the enriched terms included "Chemical Synaptic Transmission", "Potassium Ion Transmembrane Transport", and "Positive Regulation of Potassium Ion Transmembrane Transporter Activity" (Supplementary Fig. 5b, Supplementary Data 4). Additionally, Gene Set Enrichment Analysis (GSEA) comparing nerve crush to naïve conditions revealed that the top three upregulated pathways were "Axon Development", "Positive Regulation of Programmed Cell Death", and "Positive Regulation of Leukocyte Migration", while no pathways were significantly downregulated (Supplementary Fig. 5c, Supplementary Data 5).

These analyses highlight RNase4-associated gene networks linked to injury responses, suggesting RNase4 may be involved in processes related to tissue repair and immune modulation after nerve injury.

To further assess the role of RNase4 in nociceptor function in a neuropathic pain model, we employed the sciatic nerve crush injury model. Sciatic nerve crush is a well-established model of Wallerian degeneration, characterized by axonal degeneration and myelin breakdown distal to the lesion, and is widely used to study mechanisms of neuropathic pain. Following nerve injury, cKO-BR and cKO-PR mice, along with their control littermates, exhibited a temporary reduction in mechanical or thermal sensation by 7 days after nerve injury, followed by recovery of sensory function. Von Frey testing revealed that both cKO-BR, cKO-PR (Fig. 6f, i), and control mice displayed the expected reduction in mechanical sensitivity following sciatic nerve crush, followed by a progressive recovery over time. In both knockout lines, recovery of mechanical thresholds occurred faster than in littermate controls, indicating that loss of RNase4 facilitates restoration of tactile sensitivity after injury. In contrast, Hargreaves testing of thermal sensitivity showed an opposite trend between genotypes cKOs at 14 days post-injury, with cKO-BR mice displaying a delay in thermal recovery compared to controls, while cKO-PR fully recovered. This divergence suggests that RNase4 modulates regeneration of mechanical and heat pathways in a differential, modality-specific manner, and highlights that accelerated recovery of tactile function in

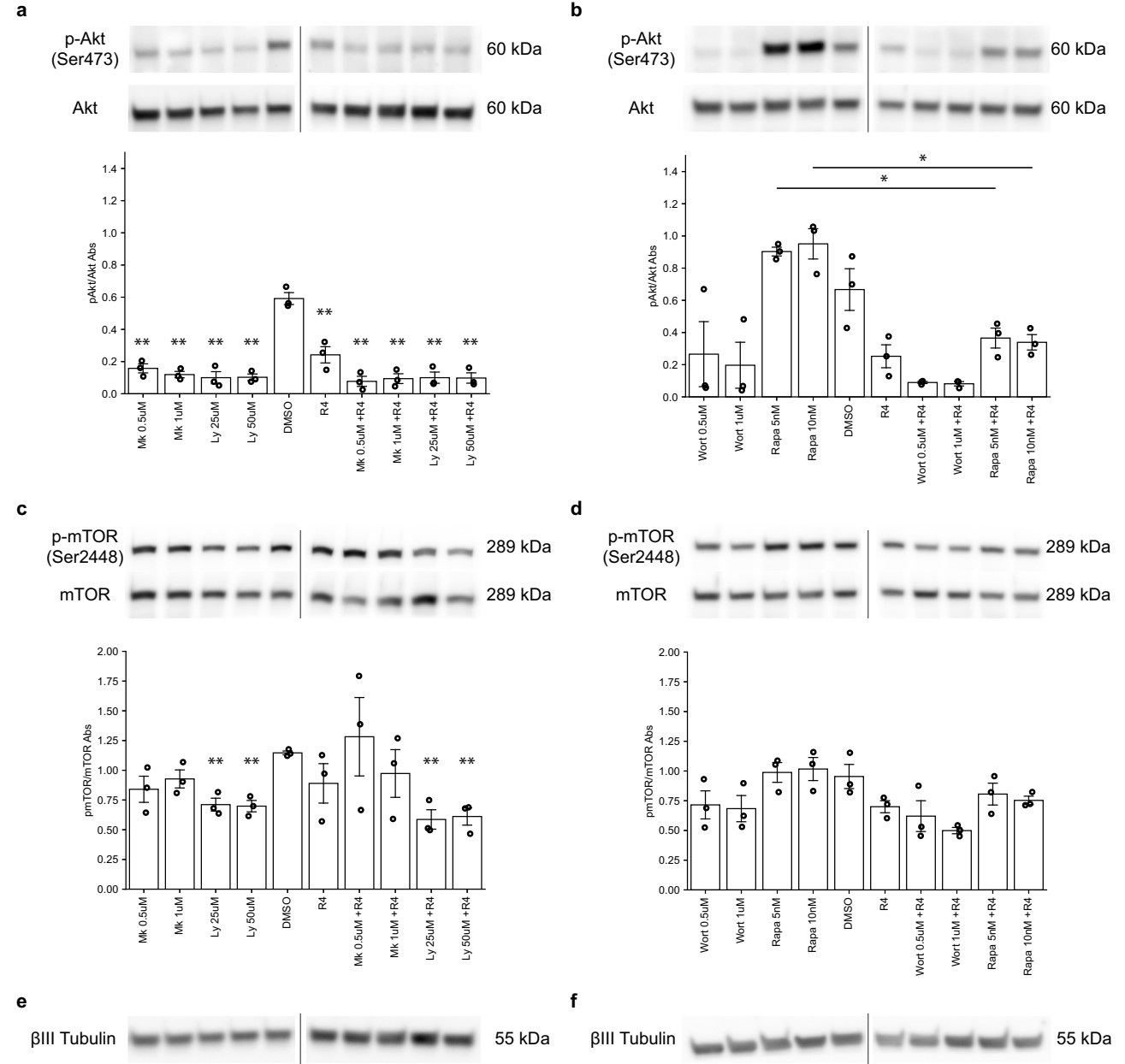

**Fig. 5 | Effect of exogenous RNase4 on the Akt-mTOR pathway in ND7/23 cells.**
**a**, **b** Representative pAkt and Akt blot for each inhibitor and RNase4 (R4) condition used in culture with bar plot of mean ± SEM showing pAkt/Akt ratio quantified by western blot with $n = 3$ independent biological replicates. Between DMSO and R4 $p = 0.00727$, whereas between DMSO and all other conditions, denoted as significant $p = 0.00119$. For Rapamycin 5 nM and Rapamycin 5 nM +R4, $p = 0.0109$ and for Rapamycin 10 nM and Rapamycin 10 nM +R4, $p = 0.0176$. **c**, **d** Representative pmTOR and mTOR blot for each inhibitor and RNase4 condition used in culture with bar plot of mean ± SEM showing pmTOR/mTOR ratio quantified by western blot with n = 3 independent biological replicates. Between DMSO and LY294002, as well as LY294002 + R4 conditions, $p = 0.00823$. **e**, **f** Representative βIII tubulin blot that was run from the same gel with each phospho/total protein pair and used as

loading control to normalize densitometry results prior to ratio calculations for each of the $n = 3$ biological replicates. Thin black lines indicate the splicing of originally non-adjacent lanes from the same blot. Inhibitors were added for 1 hour at 37 °C and include MK-2206 (Mk), LY294002 (Ly), Wortmannin (Wort), and Rapamycin (Rapa). DMSO was included as a vehicle, and 1 mg/mL RNase4 (R4) was added for 5 min after inhibitor treatment for the indicated conditions. Statistical significance between groups was determined using a two-tailed independent t-test, and the multiple hypothesis testing was corrected with Benjamini-Hochberg. Statistical significance indicated above each bar was obtained from comparisons to the DMSO group, and comparisons between RNase4 and inhibitor+RNase4 groups, as well as inhibitor and inhibitor+R4 groups, were also considered. (*$p < 0.05$, **$p < 0.01$). Source data are provided as a Source Data file.

the knockouts does not necessarily generalize to thermal nociception at all stages of the regeneration process.

**Evaluation of the role of RNase4 on repair and remyelination**
Our analysis revealed that RNase4 expression in the peripheral nervous system is predominantly restricted to unmyelinated fibers, as

observed in dorsal root ganglia (DRG; Fig. 1a–c), trigeminal ganglia (TG; Fig. 7a), and type II neurons of the spiral ganglion (Fig. 7b). These findings prompted us to investigate whether RNase4 contributes to unmyelinated fiber sorting or facilitates interactions between C-fibers and Schwann cells under normal and injury conditions.

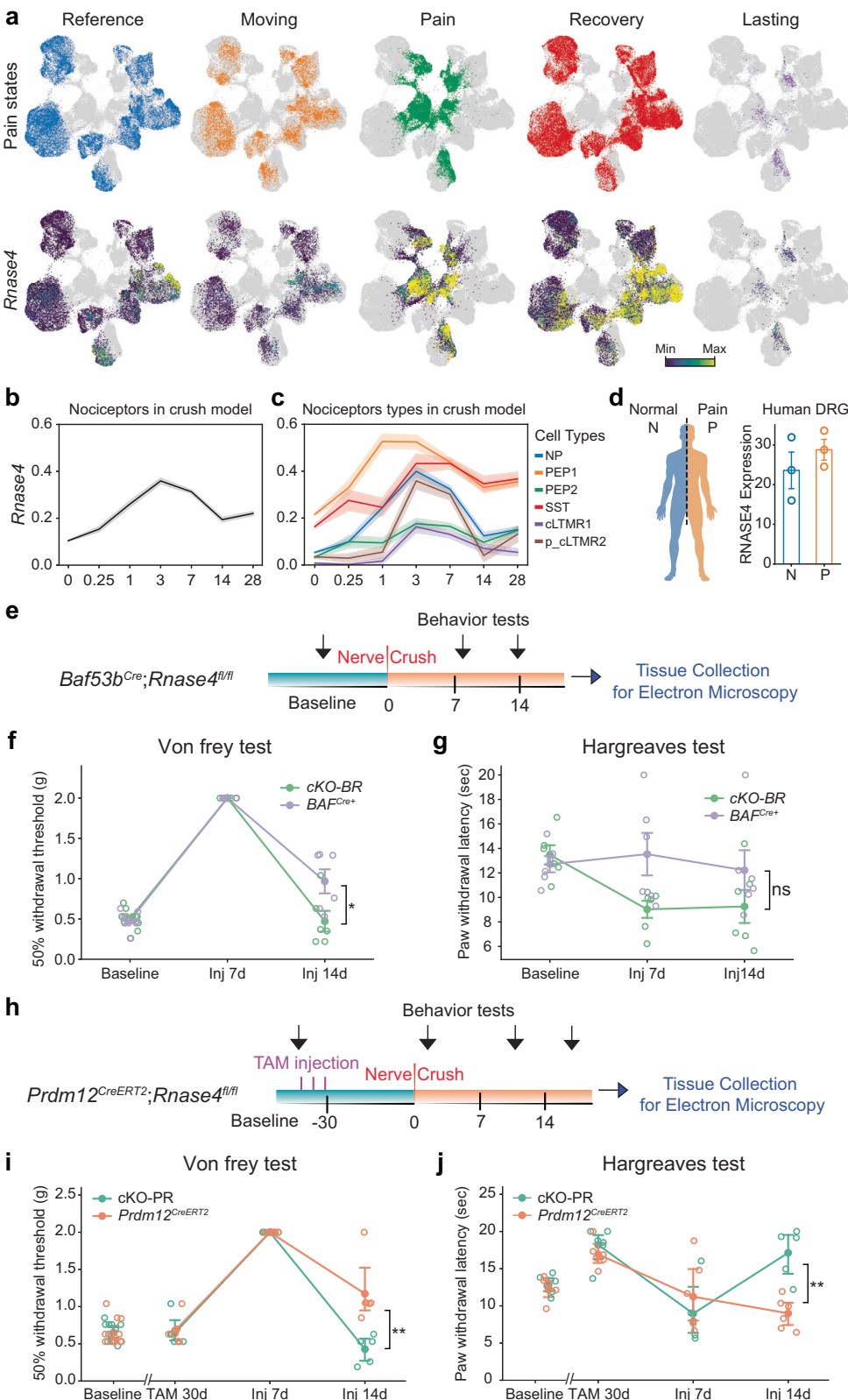

To address this, we collected sciatic nerve samples from control and *Rnase4* cKO mice and processed them for electron microscopy (EM) to assess myelin ultrastructure and axonal morphology at ultra-structural resolution (Fig. 7c–o). Mice underwent unilateral sciatic nerve crush injury (ipsilateral, Ipsi) to examine Wallerian degeneration, demyelination, and remyelination processes.

In uninjured nerves, neither G-ratio (calculated per animal; Fig. 7d), axon diameter (Fig. 7e), nor axon density (axons/mm²; Fig. 7f) differed significantly between genotypes. However, in injured nerves, *Rnase4* cKO mice exhibited a significant increase in axon density compared to controls (Fig. 7f), while the number of myelinated axons remained comparable between genotypes (Fig. 7g). These data

**Fig. 6 | RNase4 expression changes in nociceptive lineage cells following nerve crush injury and pain sensitivity in conditional RNase4 knockout mice. a** Top: UMAP plots of nociceptive lineage cells colored by different pain dynamic states. Bottom: UMAP plots showing *RNase4* expression within nociceptive lineage cells across various pain states. **b** Line plot showing average *RNase4* expression in nociceptors from the crush injury model, with SEM indicated by the shaded area. **c** Line plots representing mean ± SEM of *RNase4* expression in each nociceptor subtype in the crush injury model. **d** A bar plot representing the mean with SEM error bar of *RNASE4* expression in human DRG patients of normal (N) and pain (P) conditions. **e** Experimental timeline for *BafS3b^Cre^;Rnase4^fl/fl^* mouse line. **f** Line plot of mean ± SEM showing Von Frey test scores for *BafS3b^Cre^* control and cKO-BR mice before and after injury ($p = 0.0364$). **g** Line plot of mean ± SEM showing Hargreaves test scores for *BafS3b^Cre^* control and cKO-BR mice before and after injury ($p = 0.24$). **h** Experimental timeline for *Prdm12^CreERT2^;Rnase4^fl/fl^* mouse line. **i** Line plot of mean ± SEM showing Von Frey test scores for *Prdm12^CreERT2^* control and cKO-PR mice before and after injury ($p = 0.00741$). **j** Line plot of mean ± SEM showing Hargreaves test scores for *Prdm12^CreERT2^* control and cKO-PR mice before and after injury ($p = 0.00433$). Each group consisted of 6 mice, evenly divided by sex, with 3 males and 3 females. Statistical tests were performed with a two-tailed independent Mann-Whitney U test for between-line comparisons at the same timepoint (ns = not significant, *$p < 0.05$, **$p < 0.01$). Source data are provided as a Source Data file.

indicate enhanced axon regeneration but no alteration in myelination rates in cKO-PR mice.

Notably, the incidence of aberrant myelin structures, including focal hypermyelination, myelin infolds, and degenerating myelin, was significantly elevated in *Rnase4* cKO nerves compared to controls (Fig. 7h, i). This increase in myelin misfolding was predominantly observed in intermediate-diameter fibers, with both genotypes showing similar distributions of myelin defects (Fig. 7i). Analysis of Remak bundles in uninjured sciatic nerves revealed no differences between genotypes in bundle number or axons per bundle (Fig. 7j–l). Post-injury, cKO-PR mice exhibit similar C-fiber densities per Remak bundle comparable to uninjured levels, indicating accelerated axonal regeneration, whereas control nerves showed persistent reductions consistent with slower recovery post-injury (Fig. 7l).

cKO-PR mice also exhibited significantly reduced myelin debris enclosed by immune cells at 14 days post-injury (Fig. 7m–o), correlating with more advanced axonal regrowth. Efficient myelin debris clearance by macrophages and Schwann cells is a critical determinant of regeneration success. Schwann cell vacuolization, a hallmark of cellular stress, was observed exclusively in *Rnase4* cKO nerves (Fig. 7h).

Importantly, western blot analysis revealed increased pAkt/Akt levels upon RNase4 inhibition, indicating enhanced activation of the PI3K–AKT pathway (Fig.5). Given the established role of AKT signaling in axonal growth and regeneration[29,30] these data provide a mechanistic link between RNase4 loss and the observed acceleration of axonal regrowth. Our findings suggest that RNase4 normally restrains regenerative signaling, and that its absence promotes axonal regeneration, potentially through AKT-dependent pathways, while simultaneously perturbing Schwann cell homeostasis and myelin remodeling (Fig. 7p).

Taken together, our findings reveal a complex and unexpected role for nociceptors in nerve homeostasis and regeneration of surrounding nerve fibers (Fig. 7p, scheme). Under basal conditions, nociceptors expressing RNase4 exert control over myelination, Remak bundle formation, and fiber wrapping in the sciatic nerve. Following injury, these nociceptors influence axon regeneration and remyelination during Wallerian degeneration. This dual effect, mediated through RNase4, suggests that nociceptors act as critical modulators of myelin dynamics, likely by influencing Schwann cell activity and macrophage behavior. This dual mechanism highlights the complex role of nociceptors and RNase4 in maintaining nerve tissue equilibrium.

These results not only expand our understanding of nociceptor function in the peripheral nervous system beyond pain sensation but also hint at the potential of targeting RNase4 in nociceptors as a therapeutic strategy for promoting nerve repair and regeneration in pathological conditions.

## Discussion

The RNase A family of proteins has diverse biological functions and is implicated in the development of various diseases, immune dysfunction, neurodegenerative conditions, and cardiovascular disorders. In this study, we identify RNase4 as a marker gene for small-diameter, unmyelinated nociceptors. It plays a role in maintaining their normal function, supporting the morphology of neighboring neurons, including axon diameter and myelin structure, and modulating pain perception. Interestingly, the absence of RNase4 was linked to faster mechanical recovery and nerve regeneration after injury. These findings highlight RNase4's role in nociceptor homeostasis in adults and suggest its potential for timely recovery after peripheral nerve injury.

### RNase4 and nerve injury

In the nervous system, RNase4 has been explored in the context of neuroprotection, with few studies indicating its role in promoting neuronal survival, angiogenesis, and neurogenesis under stress[11]. Some studies have also suggested a potential link between RNase4 and neurodegenerative diseases, including amyotrophic lateral sclerosis (ALS)[11] and Huntington's disease[31], though these associations are still emerging and require further investigation[11]. RNase4 was shown to exhibit angiogenic activity. Angiogenesis plays a vital role in nerve regeneration following nerve injury-induced Wallerian degeneration by providing metabolic support, facilitating immune-mediated debris clearance, and guiding Schwann cells and regenerating axons. Our findings suggest that nociceptor-derived RNase4 may contribute to this process. In our sciatic nerve injury model, *Rnase4* deletion enhances debris clearance, as evidenced by a reduction in myelin debris and fewer infiltrating macrophages – key mediators of myelin debris removal – within the distal nerve two weeks post-injury. Efficient myelin clearance is essential for promoting nerve regeneration and remyelination, and we observed accelerated axon regeneration, enhanced remyelination, and faster sensory recovery in *Rnase4*-deficient mice. While further studies are needed to elucidate the mechanism by which RNase4 modulates neuroinflammatory response, it is conceivable that while rapid axon regeneration could initially appear beneficial for functional recovery, it may also promote maladaptive growth with potential long-term repercussions. It is therefore possible that RNase4 plays a role in coordinating the appropriate regenerative processes to achieve the best outcome.

### RNase4 and Neuronal function

The expression of RNase4 specifically in nociceptors also raises the possibility that it plays a cell-autonomous role in regulating the molecular pathways that control nociceptor excitability, which is crucial for pain sensation. Our study reveals that RNase4 contributes to the expression of specific ion channel components in nociceptors. Notably, *Kcnf1* (Kv5.1) and *Kcng3* (Kv6.1), which do not independently form ion channels, interact with Kv2 (Kv2.1/2.2) to assemble functional tetramers, significantly altering the biophysical properties of Kv2. Kv2 is a key modulator of somatosensory neuron electrical activity[24,32,33], and its dysfunction following peripheral nerve injury has been linked to increased neuronal excitability, facilitating a neuropathic pain state[32]. Although no overt pain phenotype was

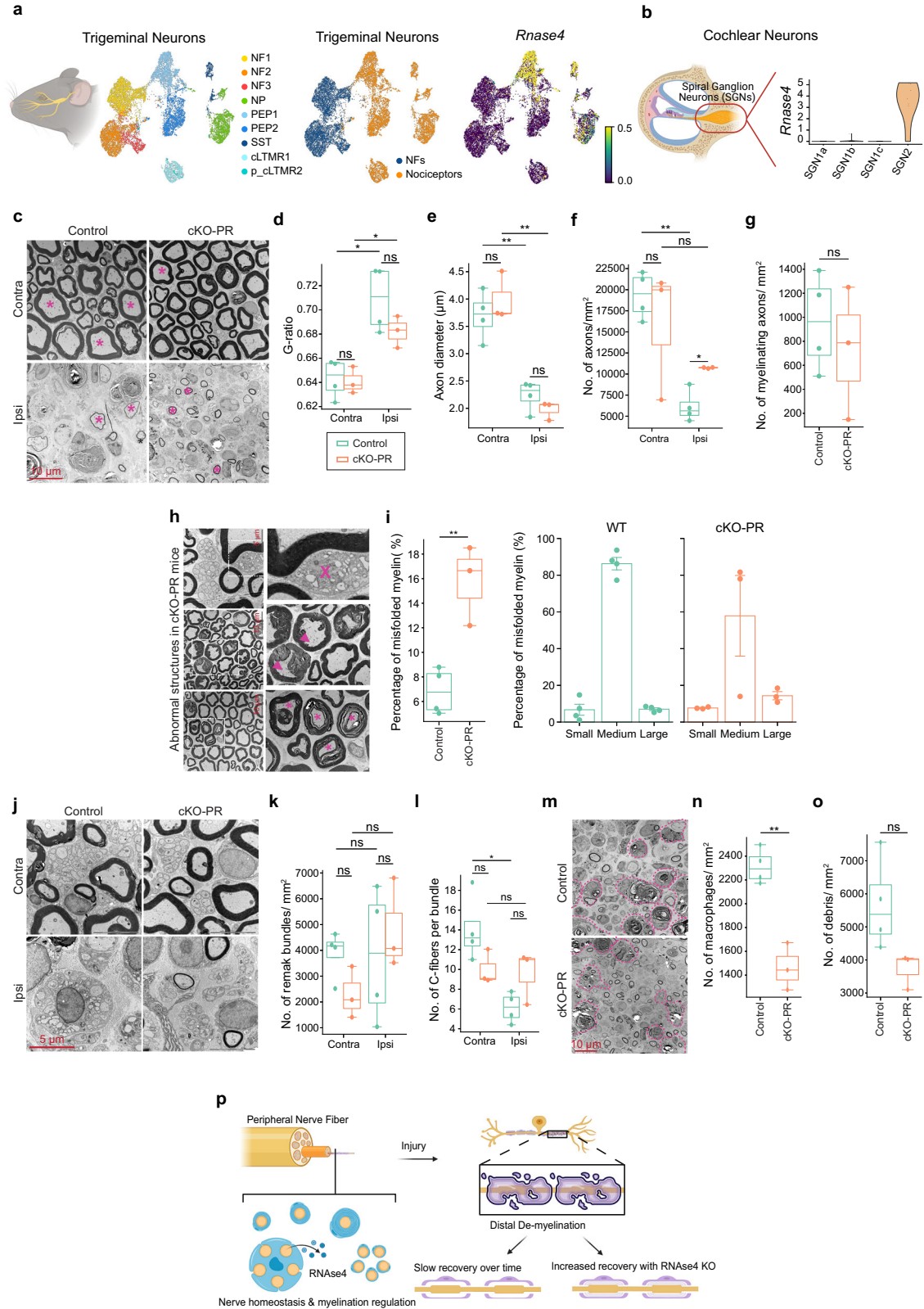

observed under normal conditions in the absence of RNase4 when deleted during embryogenesis, a notable mechanical allodynia emerged several days after the acute deletion of RNase4 in adult mice. This phenotype was transient, resolving within a month, suggesting that nociceptors activate compensatory mechanisms to restore homeostasis. RNase5 (angiogenin), a structurally related

RNase A superfamily member with established roles in neuroprotection[34,35], was undetectable in sensory neurons (Supplementary Fig. 6) under basal condition or in *Rnase4* cKO mice, excluding simple RNase superfamily compensation. This temporal divergence suggests nociceptors activate compensatory mechanisms during development–but not acutely in adults–to restore

**Fig. 7 | Non-cell-autonomous effects of *RNase4* removal in nociceptors: impact on axon diameter, myelin formation, and injury response. a** UMAP of trigeminal ganglion (TG) neurons showing neuronal subtypes and RNase4 expression. **b** RNase4 expression in cochlear spiral ganglion neurons (SGN), enriched in the SGN2 population. **c** Representative transmission electron microscopy (TEM) images of transverse sciatic nerve sections from *Prdm12*[CreERT2] (Control, *n* = 4) and *Prdm12*[CreERT2]; *Rnase4*[fl/fl] (cKO-PR, *n* = 3) mice, 14 days after sciatic nerve crush injury. Scale bar, 10 μm. Quantification of g-ratio (**d**), axon diameter (**e**), and axon density (**f**) in contralateral (Contra) and ipsilateral (Ipsi) sciatic nerves. Axon counts: Control, *n* = 2691 (Contra) and *n* = 1130 (Ipsi); cKO-PR, *n* = 2024 (Contra) and *n* = 904 (Ipsi). **g** Density of myelinated axons in contralateral distal sciatic nerves (Control, *n* = 24 images; cKO-PR, *n* = 16 images). **h** TEM images highlighting Schwann cell vacuolization (X), focal hypermyelination (arrowheads), and abnormal myelin structures including intermyelin infolds (asterisks) in cKO-PR nerves. Scale bar, 10 μm. **i** Quantification of misfolded myelin as a percentage of total myelin (Control, *n* = 2067; cKO-PR, *n* = 2623), and stratified by axon diameter class (Small, <2 μm;

Medium, 2–8 μm; Large, >8 μm). Analysis of non-myelinated Remak bundles showing representative TEM images (**j**), bundle density (**k**; Control, *n* = 24 images; cKO-PR, *n* = 16 images), and number of axons per bundle (**l**; Control, *n* = 81 Contra and *n* = 90 Ipsi bundles; cKO-PR, n = 90 Contra and n = 40 Ipsi bundles). Scale bar, 5 μm. TEM images (**m**) and quantification of macrophage number (**n**) and myelin debris (**o**) in ipsilateral sciatic nerves. Macrophages are outlined with pink dashed lines; asterisks indicate myelin debris engulfed by macrophages or Schwann cells. **p** Model summarizing RNase4-dependent regulation of peripheral nerve homeostasis and repair following injury. Schematics in (**a**, **b** and **p**) were created in BioRender. Techameena, P. (2026) https://BioRender.com/j2vwxbp; and Techameena, P. (2026) https://BioRender.com/1fj082y. Data are presented as median ± SEM. Statistical comparisons were performed using two-tailed t-tests with Benjamini–Hochberg correction (ns, not significant; **$p < 0.01$; ***$p < 0.001$). Box plots indicate median (center line), interquartile range (box), and minima/maxima (whiskers). Source data and exact *p*-values are provided as a Source Data file.

homeostasis without RNase4. These findings underscore RNase4's specific importance in maintaining adult nociceptor signaling and sensory thresholds.

## RNase4, p53 pathway, and senescence

One of our recent studies has additionally highlighted a link between the senescence of nociceptors, the development of neuropathic pain, and the activation of p53[13]. In this context, RNase4 has also been recognized as a senescence-associated super enhancer[36] in its ability to suppress p53-mediated apoptosis[36], where its depletion in irradiation induced senescent fibroblasts significantly reduced their viability. The top GO term associated with RNase4 expression in naïve DRG was "Regulation of Signal Transduction by P53 Class Mediator" (Supplementary Fig. 1a), hinting at a potential link to senescence pathways. However, direct effects of RNase4 on nociceptor senescence remain to be investigated. Since it is now known that targeting senescence in neuropathic pain aids its recovery, it is also plausible that the beneficial effects of removing RNase4 outlined here could, at least partially, be exerted through this mechanism. In our current study, we propose that by maintaining cellular homeostasis both under normal and stressed conditions, RNase4 supports nociceptor function following injury. However, the therapeutic potential of targeting RNase4 may be constrained, as not all nociceptors express RNase4 at the transcriptomic level (in our transcriptomic dataset). Consequently, any therapeutic benefit could be partial unless RNase4 can be released by nociceptors and taken up by neighboring cells, although this remains a hypothesis that requires further investigation. Structural evidence from inter-lineage communication suggests that RNase4 released by nociceptors may influence the axon diameter and myelination of larger sensory neurons. If these mechanisms are confirmed, RNase4-targeted therapies could represent a promising strategy for reversing nociceptor-associated senescence and improving pain management, particularly in the context of neurodegenerative conditions or nerve injury. While intriguing, the therapeutic implications of this finding remain to be tested.

Nociceptors are uniquely positioned at the frontlines of the peripheral nervous system, acting as the body's earliest responders to injury or harmful stimuli. Our study shows that those specialized sensory neurons that detect noxious signals serve not only to initiate pain perception but also to safeguard the integrity of the entire nerve network. In physiological conditions or when tissue damage or stress is detected, nociceptors react by releasing a cascade of bioactive molecules, including RNase4, which may play a pivotal role in modulating local tissue homeostasis and neuronal activity. This immediate release of nociceptive signaling molecules helps trigger defensive responses, including inflammation and neuroplastic changes, which collectively work to protect the nerve from further harm. By adjusting their own excitability and influencing the surrounding cellular environment, nociceptors regulate key processes like axonal repair, Schwann cell

activity, and even immune cell recruitment, thus maintaining nerve function under duress. Their ability to balance sensitivity to noxious stimuli with repair and protective mechanisms places nociceptors at the heart of nerve homeostasis, making them indispensable not only for pain detection but for the broader preservation of neural homeostasis in response to damage.

## Methods

### Mice and ethics

Adult male and female wildtype and transgenic mice of the C57BL/6 J background (aged 2–4 months) and embryos (embryonic day 15.5–17.5) were used in this study. The animals were housed under a 12-hour light–dark cycle at 24 °C, with ad libitum access to food and water. All animal care and experimental procedures were permitted by the Ethical Committee on Animal Experiments (Stockholm North committee) and conducted according to The Swedish Animal Agency's Provisions and Guidelines for Animals Experimentation recommendations with ethical permit number 17396-2022.

### Mice construction. EasyCRISPR

We developed two mouse models in this study, *Rnase4*[fl/fl] and *Prdm12*[CreERT2], both using the Easi-CRISPR approach[21,37]. C57BL/6J mouse strain (Jax strain 000664) was used as zygote donor, and the transgenic mouse experiments were performed as described previously[37,38].

*Rnase4*[fl/fl] mouse: The Rnase4 gene (ENSMUSG00000021876) has only one coding exon (ENSMUST00000022428). Creating a LoxP flanked (floxed) allele for single coding-exon genes would be challenging because inserting LoxPs may disrupt regulatory regions such as promoters and regulatory sequences in 5' and 3' ends, respectively. We used a split exon strategy to insert an 'artificial intron with LoxP sites flanking a branching point' at the split site to create a conditional knockout allele for RNAse4[20,39]. One of the limitations of this approach is the unpredictable effect of splitting the exon (and the inserted cassette) on the normal expression of the conditional allele. Reasons for this could be many. Two important ones are: (a) the incomplete splicing out of the artificial intron (before Cre) and (b) the unknown effect on the normal expression of the transcript due to the presence of unnatural sequences within the exon. Because of this, the conditional alleles can diminish the gene expression (hypomorphic) or result in a complete loss of expression even before Cre-mediated excision of the branching point[39]. Using RNAscope technology, *RNase4* mRNA expression levels were comparable between the *RNase4*[fl/fl] mice and their littermates (those without artificial intron insertion in the *RNase4* gene), while breeding with Cre mice or with CreERT2 mice followed by tamoxifen injection led to a decrease in the *Rnase4* mRNA expression. Therefore, we presume that the split exon strategy for the *Rnase4* locus did not affect the normal expression of the gene. Details

of the *Rnase4* SCON strategy and generating the mouse model are given in Supplementary Data 6.

The *Prdm12*<sup>CreERT2</sup> mouse was generated by deleting the coding regions of the first two exons of *Prdm12* and by inserting CreERT2-SV40 polyA cassette in place of it using the *Easi*-CRISPR approach. Details of the *Prdm12*<sup>CreERT2</sup> mouse model design are given in the supplementary file 2. The *Prdm12*<sup>CreERT2</sup> is a KI/KO mouse model. It can be used as a Cre driver mouse when it is used as a heterozygous and a *Prdm12* knockout mouse when the allele is bred to homozygous or as a Cre driver mouse when it is used as a heterozygous. The knockout phenotype of the mouse is not characterized in this study. Details and sequences used to generate the mouse line is given in Supplementary Data 7.

### Adult mice and embryos

*Prdm12*<sup>CreERT2</sup> mice were crossed with Ai14 (*Rosa26*<sup>tdTOM</sup>, The Jackson Laboratory, 007914) for immunohistochemical and iDISCO staining characterization. Only heterozygote *Prdm12*<sup>CreERT2</sup> were bred to avoid the possible generation of full *Prdm12* knockouts. To characterize the effects of RNAse4 conditional knockout in neurons, the *RNAse4*<sup>fl/fl</sup> mice were crossed with *BafS3b*<sup>Cre</sup> (The Jackson Laboratory, 027826[40]) for constitutive knockout during development or *Prdm12*<sup>CreERT2</sup>; Ai14 mice for timed knockout in adulthood. Both Cre lines were mated for heterozygote pups only.

Ear biopsies for genotyping were taken by facility technicians at weaning. DNA was extracted from tissue biopsies using the Phire Animal Tissue Direct PCR Kit (Thermo Scientific, F140WH) and genotyped with PCR gel electrophoresis using the primers found in Supplementary Table 1.

### Tamoxifen treatment

On the day of injection, tamoxifen (Sigma, T5648-1G) was dissolved in corn oil (Sigma, C8267-500ML) at a concentration of 20 mg/ml.

For staining characterization of *Prdm12*<sup>CreERT2</sup>; Ai14 adult mice, tamoxifen was delivered via intraperitoneal injection (i.p.) for 3 or 5 consecutive days at a dose of 100 mg/kg, and were then allowed to recombine for 1 week. Similarly, 3 doses of 100 mg/kg tamoxifen were injected i.p. in *Prdm12*<sup>CreERT2</sup>; *Rnase4*<sup>fl/fl</sup> mice for behavior characterization according to the experiment timeline.

For characterization of *Prdm12*<sup>CreERT2</sup>; Ai14 in embryos, tamoxifen was injected i.p. in the pregnant mother at a dose of 100 mg/kg for 3 consecutive days prior to collection for E15.5–E17.5 embryos. Embryo collection was performed around 4 hours or the day after the final injection. Tail tips were also collected for genotype confirmation using the protocol and primers described above.

### Sciatic nerve crush injury

As described in a reported method[41,42], buprenorphine (0.1 mg/kg) and carprofen (5 mg/kg) were first administered subcutaneously, followed by the surgical exposure of the right sciatic nerve under isoflurane anesthesia. The nerve was then subjected to a controlled crush injury for 15 seconds using hemostatic forceps (Fine Science Instruments; 0.4 mm tip angled, Cat. No. 11063-07), followed by a second crush at the same site oriented orthogonally to the initial injury. After the procedure, the skin was sutured. Distal nerve stumps located 1 mm from the crush site were collected for electron microscopy (EM) analysis. A subsequent dose of carprofen (5 mg/kg) was administered subcutaneously 24 hours post-surgery.

### Pain behavior

Mechanical sensitivity was assessed using calibrated von Frey filaments following the up–down method described previously[43–46]. Mice were individually placed in transparent chambers positioned on a metal mesh platform (6 × 6 mm grid) and allowed to habituate for 20–60 minutes before testing. The plantar surface of the hind paw was stimulated with a series of monofilaments (Stoelting, IL, USA) spanning forces from 0.008 to 2 g. A positive response was defined as a clear paw withdrawal or shakes. Six responses were obtained for each paw per animal.

Thermal nociception was evaluated using the Hargreaves plantar test as previously reported[43,44]. Animals were acclimated for 20–60 minutes in Plexiglas chambers placed on a glass surface. A focused radiant heat source (IITC Inc., Woodland Hills, CA, USA) set at 30% intensity was directed at the central plantar region through the glass. Paw withdrawal latency was recorded, with a 20-second cutoff applied to prevent tissue injury.

### RNAscope and immunohistochemistry

Adult mice were sacrificed by isoflurane overdose followed by transcardiac perfusion with chilled PBS and then 4% PFA, where sciatic nerve, spinal cord, and lumbar DRG were collected. Following dissection, tissues of interest were post-fixed in PFA for 1 hour for immunohistochemistry (IHC) or overnight for RNAscope at 4 °C.

For embryo collection, the mother was sacrificed by $CO_2$ overdose, and embryos were dissected out on ice. To process embryos for sectioning, the ventral organ mass was removed before post-fixation in PFA for 6 hours at 4 °C.

After PFA fixation, samples were then transferred to 30% sucrose for cryoprotection overnight at 4 °C. Embryo samples were further dissected to isolate the spinal column for further processing.

Dissected tissues were then embedded in optimal cutting temperature (OCT) medium and stored in −80 °C. Blocks were sectioned using Epredia™ CryoStar™ NX70 cryostat at 14 μM and stored in −80 °C. Sections were air-dried for at least 1 hour before use.

RNA in situ hybridization was performed using RNAscope® Multiplex Fluorescent Reagent Kit v2 according to manufacturer instructions (Advanced Cell Diagnostics, ACD). Target probes, Mm-Prdm12-C1 (ACD, 524371), Mm-Kcng3-C1 (ACD, 1230601-C1), Mm-Prdm12-C2 (ACD, 524371-C2), Mm-Sst-C2 (ACD, 404631-C2), Mm-Pvalb-C3 (ACD, 421931-C3), Mm-*Rnase4*-C1 (ACD, 1041951-C1), Mm-Kcnf1-C2 (ACD, 508731-C2), Mm-Calca-C3 (ACD, 578771-C3) and Mm-tdTomato-C3 (ACD, 317041-C3D) were used according to protocol.

IHC was performed on sectioned slides or following RNAscope using the following antibodies as necessary: 1:400 TH (Pel-Freez, P40101), 1:400 TrkA (R&D Systems, AF1056), 1:250 NF200 (Millipore, AB1991), 1:250 peripherin (Abcam, ab39374), 1:200 CGRP (Immunostar, AB_572217), 1:500 rabbit anti-PV (Swant, PV27a), and 1:500 rabbit anti-RFP (Rockland, 600-401-379). Isolectin B4 conjugated with fluorescein was added at 1:100 together with primary antibodies. Secondary antibodies were used at a concentration of 1:1000 for 1 hour at room temperature: donkey anti-rabbit IgG AF 488 (Invitrogen, A-21206) or donkey anti-rabbit IgG 555 (Invitrogen, A-31572).

Slides were then counterstained with 1:200 NeuroTrace™ 435/455 Blue Fluorescent Nissl Stain (Invitrogen, N21479) or 1:1000 DAPI (Invitrogen, D1306) before sealing with DAKO fluorescent mounting medium (Agilent, S3023) and stored at 4 °C.

Images were taken using a Zeiss LSM 800 or Zeiss LSM 800 Airyscan confocal microscope and analyzed in ImageJ.

### Volume immunostaining and optical clearing of fixed mouse embryos

Embryos were collected from pregnant females euthanized by $CO_2$ inhalation and fixed overnight at 4 °C in modified Zamboni fixative (4% paraformaldehyde, 0.2% picric acid in phosphate buffer, pH 7.4). Samples were post-fixed for 1 hour at room temperature, washed, and stored in PBS containing 0.02% sodium azide at 4 °C until processing. Whole-mount immunolabeling and clearing were performed using the iDISCO+ protocol[47–49].

Genotyped E15.5 embryos were washed in 1× PBS and dehydrated in a graded methanol series (20–100%, 1 hour per step).

Autofluorescence was reduced by overnight incubation in 5% $H_2O_2$ in methanol at 4 °C. Samples were rehydrated through a reverse methanol gradient and incubated sequentially in permeabilization buffer (0.2% Triton X-100, 20% DMSO, 0.3 M glycine in PBS with 0.02% sodium azide) and blocking solution (0.2% Triton X-100, 10% DMSO, 6% normal donkey serum), each for 48 hours at 37 °C.

Primary antibodies (rabbit anti-RFP, 1:500, Rockland; goat anti-TrkA, 1:400, R&D Systems) were applied for 4 days at 37 °C in antibody buffer (0.2% Tween-20, 10 mg/ml heparin, 5% DMSO, 3% normal donkey serum in PBS with sodium azide). Following extensive washes, embryos were incubated for 4 days with Alexa Fluor–conjugated secondary antibodies (donkey anti-rabbit AF555; donkey anti-goat AF647; 1:500, Invitrogen).

After staining, tissues were dehydrated in methanol, cleared in dichloromethane/methanol (2:1) followed by 100% dichloromethane. Finally, samples were transferred into 100% dibenzyl ether and stored in this medium until imaging.

### Light sheet fluorescence microscopy and 3D image reconstruction

Cleared samples were imaged using an Ultramicroscope II (LaVision Biotec, Bielefeld, Germany) operated with Imspector™ software[49]. The system was equipped with a 2×/0.5 Olympus objective, an MVX-10 zoom body (0.63–6.3×), a 6.5 mm working-distance spherical aberration–corrected dipping cap, and an Andor Neo sCMOS camera (2560 × 2160 pixels). Excitation was provided by Coherent OBIS lasers at 488 nm, 561 nm, and 640 nm with appropriate emission filters.

Acquisitions were performed in multicolor mode using single-sided illumination. Laser power was set to 60–80% with a 100 ms exposure time. The light sheet numerical aperture was set to 0.156 with 70% sheet width. Z-stacks were collected at step sizes between 1.89 and 4.8 µm. No tiling was applied.

Image series were acquired as 16-bit TIFF stacks and converted to IMS format using Imaris File Converter (v9.7.2, Bitplane). Three-dimensional rendering and analysis were conducted in Imaris (v9.7.2). Background subtraction was applied during processing, and brightness and contrast were minimally adjusted for the figure.

### Electron microscopy

Following behavioral assessment after nerve injury experiment, animals were mice were sacrificed by isoflurane overdose and transcardially perfused with 2.5% glutaraldehyde and 4% paraformaldehyde in 0.1 M phosphate buffer[50]. A distal segment of the sciatic nerve (~1 mm distal to the crush site) was dissected and immersion-fixed for 1 hour at room temperature in 2.5% glutaraldehyde and 1% formaldehyde prepared in 0.1 M Sørensen phosphate buffer (pH 7.4), then stored at 4 °C until further processing.

Samples were rinsed in 0.1 M phosphate buffer and post-fixed in 2% osmium tetroxide in the same buffer (pH 7.4) for 2 hours at 4 °C. Tissues were dehydrated through a graded ethanol series, stained en bloc with 0.5% uranyl acetate, followed by acetone dehydration, resin infiltration, and embedding in LX-112 resin (Ladd Research).

Ultrathin sections (approximately 80–100 nm) were cut using a Leica EM UC7 ultramicrotome and mounted on formvar-coated slot grids. Sections were contrasted with uranyl acetate and lead citrate before examination on a Hitachi HT7700 transmission electron microscope operated at 80 kV. Images were captured using a 2k × 2k Veleta CCD camera (Olympus Soft Imaging Solutions).

### Image quantification

For immunohistochemistry: To examine the distribution of *Rnase4* in DRG neuron sizes, *Rnase4*⁺ neurons were identified via RNAscope in ImageJ and their cell borders determined according to NeuroTrace™ Nissl Stain. The area of *Rnase4*⁺ neurons was measured, where the diameter was calculated as the square root of the area divided by p and then multiplied by 2 (Area = p * (Radius)$^2$). A total of 4 sections taken at 20x magnification of different lumbar DRGs were quantified from 3 wildtype mice. The number of Peripherin⁺ was counted manually using the multi-point tool from ImageJ and calculated accordingly in proportion to Nissl⁺ neurons in a minimum of 3 sections from 3 wildtype mice.

To determine the recombination efficiency of *Prdm12^CreERT2^; tdTomato* (from crossing with the Ai14 Cre-dependent reporter line) and *Prdm12* were labelled by RNAscope and counted using the multi-point tool in ImageJ. The number of *tdTomato*⁺ neurons was divided by the number of *Prdm12*⁺ neurons to obtain the proportion of Cre-dependent *tdTomato* expression in *Prdm12*⁺ DRG neurons from 4 lumbar DRG sections taken at 20x magnification of 4 wildtype mice.

For electronic microscopy image analysis, g-ratio measurements were performed using the Napari software platform. Axons and myelin were segmented and selected automatically using deep learning algorithms integrated with morphological metric tools and the Napari GUI plugin. The g-ratio was calculated as the ratio of the inner axon diameter to the total diameter (axon plus myelin sheath), with diameters derived from the measured perimeters divided by π. Additional EM analyses in this study were conducted using ImageJ software. Axon counting and selection for these analyses were performed manually.

### Westernblot

ND7/23 cells (mouse neuroblastoma x rat dorsal root ganglion neurons) were obtained commercially and cultured according to protocol in high glucose DMEM media (Dutscher, 509053) containing 10% heat inactivated fetal bovine serum (FBS, Sigma, F9665), 2mM L-glutamine (Cytiva, SH30034.01), 100 U/mL penicillin and 100 µg/mL streptomycin (Cytiva, SV30010) at 37 °C, 5% $CO_2$, in a humidified chamber with media change every 2-3 days. Prior to the experiment, cells were serum-starved overnight in DMEM with 2mM L-glutamine, and then incubated for 1 hour with the presence of the following: DMSO (vehicle, Sigma), MK-2206 (Selleck Chemicals, S1078-5mg), Wortmannin (Sigma, 681676-1MG), Rapamycin (Cell Signalling Technologies, CST CST-9904S), or LY294002 (CST, CST-9901S). Mouse recombinant RNase4 was obtained from OriGene (TP501018SE) and added at a concentration of 1 µg/mL for 5 minutes after inhibitor incubation. Pellets were then collected by aspiration in the same media and stored in −80 °C until extraction. Three independent cell culture experiments were performed.

For primary culture, adult wild-type mice were anesthetized using isoflurane, and cardiac perfusion with chilled PBS was performed. DRGs were dissected in a laminar flow hood into DMEM medium (Gibco, 31966-021), followed by a 30-minute enzymatic digestion with 12.5 mg/mL collagenase IV (Sigma, C5138) at 37 °C, and another 30-minute digestion with 6.25 mg/mL collagenase IV and 2.5 mg/mL dispase II (Gibco, 17105-041). Enzyme-containing media was then removed, and media with 10% FBS was added to stop the reaction. The solution was then serially titrated with 3 progressively smaller glass Pasteur pipettes (VWR, 612-1701) refined over a Bunsen burner (Integra), and coated with 0.2% bovine serum albumin (BSA, Sigma, A4503-10G). The solution was then filtered through a 70 mM cell strainer (Corning, CLS431751), and debris was removed by centrifugation into a bottom layer of 15% BSA. The supernatant was removed, and the pellet was then resuspended in Neurobasal media (Gibco, 21103-049) supplemented with 1X B-27 plus supplement (Gibco, A35828-01), GlutaMAX supplement (Gibco, 35050-061), 5 mg/mL cytosine arabinoside (AraC, Sigma, C6645), 50 ng/mL neural growth factor (NGF, PeproTech, 450-34), 100 U/mL penicillin, and 100 µg/mL streptomycin. Cells were then seeded in pre-coated wells: 30 minutes at 37 °C in poly-L-lysine (Sigma, P4707) followed by overnight at 37 °C in recombinant human Laminin 521 (Gibco, A29248). The media was changed the following day and thereafter once every 2 days. One week after seeding, cells were stimulated with 1 mM capsaicin (Sigma, PHR1450) or an

equivalent of ethanol (vehicle) for 60 seconds at 37 °C, thereafter the cell pellet was collected for further analysis.

Protein was extracted from collected pellets using RIPA buffer (Sigma, R0278-50ML) supplemented with Halt™ protease inhibitor cocktail (ThermoFisher, 78439), phosphatase inhibitor cocktail 2 (Sigma, P5726-1ML), and phosphatase inhibitor cocktail 3 (Sigma, P0044-1ML) according to manufacturer recommendations. Quantification was performed using the Quick Start™ Bradford Protein assay (Bio-Rad, 500-0205) and measured with a NanoDrop spectrophotometer using reference standards diluted from 2 mg/mL BSA (ThermoFisher, 23209).

For Western blot, samples and Precision Plus Protein Dual Color Standards (Bio-Rad, 1610374). were processed with 10 mM dithiothreitol, and NuPAGE™ LDS Sample Buffer (ThermoFisher, NP0007). SDS-PAGE was performed using precast mini-PROTEAN TGX gels with a 4-20% gradient (Bio-Rad, 4561096) in NuPAGE™ MOPS SDS running buffer (ThermoFisher, NP0001). Transfer was performed for 1 hour with 100 V at 4 °C using 20% methanol in Tris/Glycine Buffer (Bio-Rad, 1610734) onto PVDF membranes (GE Healthcare, RPN303F). Blocking was performed for 1 hour at room temperature using 5% BSA or nonfat dry milk (Bio-Rad, 1706404) in tris-buffered saline (TBS, ThermoFisher, J62938.K2) with 0.1% Tween-20 (TBS-T).

The following antibodies were applied overnight at 4 °C at a concentration of 1:1000: rabbit anti-Axl (CST, 8661), rabbit anti-phospho-Axl Tyr702 (CST, 5724), rabbit anti-mTOR (CST, 2983), rabbit anti-phospho-mTOR Ser2448 (CST, 5536), rabbit anti-Akt (CST, 9271), rabbit anti-phospho-Akt (CST, 9272), and mouse anti-βIII tubulin (Promega, G712A). Subsequent washes were performed in 1% BSA in TBS-T or 0.5% nonfat dry milk. Secondary antibodies, HRP-linked anti-rabbit IgG (CST, 7074) or anti-mouse IgG (CST, 7076), were then applied at 1:3000 for 1 hour at room temperature. Membranes were then washed and incubated for 4 minutes with SuperSignal™ West Pico PLUS Chemiluminescent Substrate (ThermoFisher, 34577) and imaged with ImageQuant LAS 4000 (GE Healthcare). Blots were stripped with Restore™ Western Blot Stripping Buffer (ThermoFisher, 21059) for 30 minutes at room temperature, to enable the probing of phosphorylated forms, followed by the total protein forms and then the loading control (tubulin) when relevant.

Bands were then analyzed and quantified in ImageJ using the 'analyze gels' function. The area under the integrated density curve under each lane profile was taken, while the background was considered by manually drawing a horizontal straight line across the peak base. In cases where the visible artifact signal was close to the band and merged with the peak signal on one side, another vertical straight line was drawn between the band peak and the visible artifact peak. Each band area was further normalized to the tubulin control, and then the ratio of the phosphorylated form to the total protein was taken for analysis. Original full scan blots are provided in the Source Data file for those included in Fig. 5 and the Supplementary Information file for those included in Supplementary Fig. 4 (Supplementary Figs. 7, 8).

## DRG dissociation and single nuclei isolation[51–53]
Mice were deeply euthanized by isoflurane and perfused with 4% pre-cooled PBS. Sciatic nerve and L4-L6 DRGs were collected at 2 weeks after sciatic nerve crush injury, and then immediately put on dry ice. Frozen tissues stored at −80 °C before further processing.

Nuclei were isolated from frozen DRGs following the protocol established by the Allen Institute for Brain Science[54], with all procedures conducted at 4 °C. Tissue samples were mechanically homogenized in 2 ml of ice-cold lysis buffer containing 10 mM Tris (pH 8.0), 250 mM sucrose, 25 mM KCl, 5 mM MgCl₂, 0.1 mM DTT, protease inhibitor cocktail (1×), and 0.1% Triton X-100. RNase inhibitors were included as appropriate: RNasin Plus (0.2 U/μl) for Smart-seq3xpress preparations or Protector RNase Inhibitor for 10x Genomics workflows.

The homogenate was sequentially passed through 70 μm and 30 μm strainers to remove large debris. Nuclei were pelleted by centrifugation at 900 × g for 10 minutes and resuspended in 250 μl of chilled homogenization buffer. To further purify nuclei and eliminate residual debris, samples were subjected to iodixanol gradient centrifugation (25%/29%). The final nuclei pellet was resuspended in 500 μl of blocking buffer (1× PBS, 1% BSA, 0.2 U/μl RNase inhibitor) prior to fluorescence-activated nuclei sorting and downstream Smart-seq3xpress or 10x Genomics library preparation.

## Fluorescent-activated nuclei sorting and single nuclei RNA-sequencing (Smart-seq3xpress)[51–54]
To selectively isolate neuronal nuclei, the nuclei suspension was incubated on ice for 30 minutes with a PE-conjugated anti-NeuN antibody (1:500; FCMAB317PE, Merck). Nuclei were then collected by centrifugation at 400 × g for 5 minutes and resuspended in PBS supplemented with 0.1 μg/ml DAPI (Invitrogen D3571) and 0.2 U/μl RNase inhibitor.

Individual NeuN-positive and DAPI-positive nuclei were sorted into 384-well plates containing 0.3 μl of lysis reaction mix using a BD FACSAria Fusion or BD FACSAria III flow cytometer operated at 4 °C. Sorting gates were defined based on DAPI fluorescence and PE signal intensity.

Immediately after sorting, plates were briefly centrifuged and stored at −80 °C until processing. Single-nucleus RNA sequencing was subsequently performed using the Smart-seq3xpress protocol by the Rickard Sandberg laboratory at Karolinska Institutet.

## Rnase4-centered RNAseq analysis of iPain
The expression matrices and metadata of iPain atlas[13] were downloaded from CELLxGENE. The data was loaded and analyzed using Scanpy[55] package for Python3 with default parameters unless otherwise specified. For both iPain DRG and TG, neurons were selected and re-clustered. The nearest neighbor graphs were constructed using scANVI latent representation with "cosine" as a metric to calculate the neighbor distance. The UMAP projections of both tissues were computed with 0.3 as a minimum distance.

To perform differential gene expression analysis (DEA) between the nociceptors and NF neurons, cells from different cell types were binned into pseudo-bulk data by summing the gene expression of each cell type for each replicate. Genes found in fewer than 3 samples of the pseudo-bulk sample were removed. The DEA was performed using the DESeq2 pipeline, and Wald tests through pyDESeq2[16] package in Python3. P-values were adjusted for multiple hypothesis testing with Benjamini-Hochberg (BH) correction. Genes were identified to be cell type specific when their log₂ fold change was greater than 2.5, and their adjusted p-value was less than 0.05.

Genes correlated to RNase4 were identified using Spearman's rank correlation in Naïve and Crush at 14-day conditions. The p-values of the correlation test were corrected for multiple hypothesis testing with the BH correction. Genes with absolute correlation higher than 0.25 and adjusted p-values of less than 0.05 were kept. Gene set enrichment analysis (GSEA) was performed on these genes (both positively and negatively correlated) against Gene Ontology of biological process collection from 2023 with GSEApy[56] package in Python3.

## RNAseq analysis of Baf53b^Cre;Rnase4^fl/fl mouse line
The sequenced reads were aligned using the zUMIs pipeline version 2.9.7, which makes use of the STAR-SOLO[57] alignment tool, and the reference genome was mouse (mm39).

The data was loaded and analyzed using Scanpy[55] package for Python3 with default parameters unless otherwise specified. Nuclei with unique gene count lower than 3000 genes, percent mitochondrial gene higher than 2, and percent ribosomal gene higher than 2 were removed from the data for *Baf53b^Cre*; *RNase4^fl/fl* mouse line. For the

control line, nuclei with a unique gene count lower than 2000 genes, a percent mitochondrial gene higher than 2, and a percent ribosomal gene higher than 2 were removed.

After quality control, we performed the cell type annotation by using the pre-trained scANVI model from the iPain paper for the label transfer[13,22]. More specifically, we online updated the pre-trained model using the data from this study as a query. The query model was trained with 400 epochs, and the parameter "n_samples_per_label" was equal to 30.

DEA between the nociceptors from *BafS3b*[Cre]; *Rnase4*[fl/fl] and control lines, cells from different lines were binned into pseudo-bulk data by summing the gene expression. Genes found in fewer than 3 samples of the pseudo-bulk sample were removed. The DEA was performed using DESeq2[58] pipeline, and Wald tests through the pyDESeq2 package in Python3. P-values were adjusted for multiple hypothesis testing with the (BH) correction. Genes were identified as DE between the lines when their log$_2$ fold change was greater than 0.5, and their adjusted p-value was less than 0.05. Gene set enrichment analysis (GSEA) was performed on top 15 up and down DE genes against Gene Ontology of biological process collection from 2023 with GSEAPy[56] package in Python3.

### RNAseq analysis of Prdm12[CreERT2];Rnase4[fl/fl] mouse line

Libraries were sequenced on a NovaSeq 6000 platform (NovaSeq Control Software v1.7.5; RTA v3.4.4) using an S2 flow cell under the NovaSeq Standard workflow. The sequencing configuration consisted of 28 bp for Read 1, 10 bp for Index 1, 10 bp for Index 2, and 90 bp for Read 2.

Base call files were converted to FASTQ format using bcl2fastq (v2.20.0.422, Illumina/CASAVA suite), with quality scores encoded in Phred33 (Sanger/Illumina 1.8+ format). FASTQ files were processed using the Cell Ranger pipeline (v9.0.1, 10x Genomics), aligning reads to the mouse reference genome (mm10).

The expression matrices were loaded and analyzed using Scanpy[55] and concatenated into a single matrix. Nuclei with a unique number of genes greater than 6000 and a percentage of mitochondrial counts greater than 0.5 were filtered out. The presence of ambient mRNAs was corrected using SoupX[59]. After the correction, the matrix was normalized and log-transformed using the normalize_total and log1p functions from Scanpy, respectively. We then identified the top 3000 highly variable genes using Seurate version 3 method on the corrected count matrix from SoupX. UMAP was computed after the construction of the nearest neighbor graph. Cell type annotation was done using the label transfer method from our pre-trained scANVI model after updating the weights of the model with the current dataset.

Cell types corresponding to the nociceptive lineage were selected for further analysis. UMAP was recomputed using the min_dist=0.3. We perform DEA at the conditional level using the rank_genes_groups function from Scanpy to compare cKO-PR to WT nociceptors, and the testing method was one-tail t-test with overestimated variance of each group. For the DEA of each nociceptive subtype comparing cKO-PR to WT groups, we created a pseudo-bulk with 5 pseudo-replicates for each subtype and condition to apply pyDESeq2[16] pipeline. P-values were adjusted for multiple hypothesis testing with BH correction. Genes were identified as DE between the lines when their log2 fold change was greater than 0.5, and their adjusted p-value was less than 0.25.

### Statistical analysis

For immunohistochemistry and EM, the statistical analysis and tests were done in Python3 with "stats" module from the SciPy v1.10.1 package. Data are presented as the mean ± SEM or median ± SEM. Significant changes were identified using a two-tailed t-test, and multiple hypothesis testing was corrected with Benjamini-Hochberg.

For single-cell data analysis, Python3 with "stats" module from the SciPy v1.10.1 package was utilized. The data were imported as a DataFrame or Array respectively, through Pandas v2.0.3 or Numpy v1.22.4 package. In general, the t-test was applied when the data followed a normal distribution. Otherwise, the Mann-Whitney U test was performed. Paired t-tests were used for comparison of paired data from behavior experiments, whereas an independent t-test was used on unpaired data.

### Reporting summary

Further information on research design is available in the Nature Portfolio Reporting Summary linked to this article.

### Data availability

The sequencing data generated from this study have been deposited in the Sequence Read Archive (SRA) and Gene Expression Omnibus databases under accession code SRR27563622 (Smart-seq3xpress) and GSE318231 (10x Genomics). The processed count matrix for Smart-seq3xpress was supplied as supplemetary file in.h5ad format, and this data can be accessed from Figshare by using the following link https://doi.org/10.6084/m9.figshare.28391387. Source data are provided with this paper.

### Code availability

The code for the core analysis of this work can be found using this link https://github.com/PrachTecha/Hadjab_RNase4.

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

## Acknowledgements

The authors are grateful to Roland Baumgartner and Fatima Memic for sharing RNAscope probes. The authors would like to acknowledge the BIC facility, the BFC facility, the Animal Behavior Core Facility at Karolinska Institutet, and the Electron Microscopy Core Facility (EMil). The authors acknowledge support from the National Genomics Infrastructure in Stockholm, funded by Science for Life Laboratory, the Knut and Alice Wallenberg Foundation, and the Swedish Research Council, and SNIC/Uppsala Multidisciplinary Center for Advanced Computational Science for assistance with massively parallel sequencing and access to the UPPMAX computational infrastructure. The computations and data handling were enabled by resources provided by the National Academic Infrastructure for Supercomputing in Sweden (NAISS), partially funded by the Swedish Research Council through grant agreement no. 2022-06725. S.H., P.T., and K.Z. are supported by the KID funding at Karolinska Institutet. P.T. is further supported by the IBSA Foundation Fellowship for Scientific Research in Pain Medicine, Rheumatology, and Orthopedics. S.H. and X.F. are supported by the Wenner-Gren Foundation. C.B.G. acknowledges the grant support from the NHGRI (award number R35HG010719). We thank Szu-Hsien Sam Wu and Bon-Kyoung Koo for help with designing the Rnase4 conditional knockout allele using their SCON strategy. I.A. and S.H. were supported by the StratNeuro collaborative grant (S.H. Main applicant). S.H. was further supported by Tysta Skolan, Swedish Research Council, Hjärnfonden, and Olle Engkvist Foundation.

## Author contributions

S.H. conceived and supervised the study. R.M.Q. and C.B.G. generated the mouse line using Easi-CRISPR for this project. X.F., K.Z., P.T., and S.H. designed the experiments. X.F., K.Z., A.M., S.B., and O.G.B. performed histological staining and analysis. X.F. and K.Z. performed behavioral experiments and sample collection. K.Z. performed cell culture and western blot analysis. P.T. performed computational analyses. X.F. performed transmission electron microscopy analyses. K.Z. and C.A. performed iDISCO experiments. X.F., K.Z., P.T., R.M.Q., C.A., A.M., S.B., O.G.B., F.L., I.A., C.B.G, and S.H. analyzed and interpreted the data. P.T., draft the figures. X.F., K.Z., P.T., and S.H. wrote the original draft manuscript with input from all authors.

## Funding

## Competing interests

The authors declare no competing interests.
