## [Transparent Peer Review file · Nature Communications]

Ribonuclease 4 Functions in Nociceptor-Mediated Nerve Homeostasis

Corresponding Author: Dr Saida Hadjab

Version 0:

Reviewer comments:

Reviewer #1

(Remarks to the Author)

This manuscript identifies RNase4 as selectively expressed in unmyelinated nociceptors and reveals its dual role in nociceptor function and nerve homeostasis. Through an integrated approach combining single-cell transcriptomics, genetic mouse models, and functional assays, the authors demonstrate that RNase4 regulates both nociceptor molecular identity and influences neighboring myelinated axons. While conditional deletion of Rnase4 during development alters gene expression profiles without affecting baseline pain sensitivity, adult-onset deletion causes transient mechanical hypersensitivity. Following nerve injury, Rnase4 deletion accelerates functional recovery and remyelination. The findings establish RNase4 as an important regulator of nociceptor biology with both cell-autonomous and non-cell autonomous effects. However, the molecular mechanisms underlying these effects remain inadequately explored.

Specific comments:

1. The authors should perform co-localization studies using established nociceptor markers with RNase4 RNA FISH. Additionally, protein-level detection of RNase4 would strengthen the findings, as protein is the functional entity.
2. In Fig 1g, comparative expression data showing differential expression between neuronal subtypes or tissues would provide better context for RNase4 specificity.
3. Fig 2b and 2c rely on Nissl staining, which labels all neurons. Using DRG neuron-specific markers would provide better characterization of the cells expressing RNase4.
4. The authors highlight differentially expressed genes in Figures 2e-f but fail to present comprehensive functional enrichment analysis of these genes. The manuscript mentions alterations in zinc finger proteins, transcription factors, and immune response pathways, but these findings are not adequately presented in figures or supplementary data.
5. In Fig 2g, the nature of individual data points is unclear, and there appears to be inconsistency in the number of data points between left and right panels.
6. The authors should examine whether RNase4 expression changes during the course of behavioral testing (von Frey and Hargreaves tests), which would help connect molecular changes with behavioral outcomes.
7. Regarding the observed thicker myelin and smaller axon diameter in contralateral nerves of cKO-PR mice (Fig. 5c-f), it is important to clarify whether these are developmental effects or consequences of adult Rnase4 deletion. If the latter, the time course of these structural changes following deletion should be determined.
8. The authors propose a compensatory relationship between RNase4 and RNase5 (angiogenin) in nerve regeneration but provide no experimental evidence. Examining RNase5 expression in Rnase4 knockout conditions would substantiate this hypothesis.
9. As RNase4 is described as a secreted protein, rescue experiments administering recombinant RNase4 would demonstrate whether it has direct protective effects and strengthen the causal relationship between RNase4 and the observed phenotypes.
10. The study lacks mechanistic explanation: Does RNase4 function through its enzymatic activity or through other molecular pathways? Is it modulating known signaling pathways such as AXL?

Reviewer #2

(Remarks to the Author)

RNase4 is known to be involved in a broad range of biological processes, including the regulation of intra- or extra-cellular RNA metabolism as well as antiviral, antibacterial, and antifungal activities, neurogenesis, angiogenesis, immune modulation and host defence, is selectively expressed in neurons of the nociceptive lineage. In this manuscript, the authors

report that RNase4 is selectively expressed in sensory ganglia in the unmyelinated nociceptive lineage. They analysed the consequences of its loss of function using both a conditional KO and an inducible cKO strategy with a novel Prdm12CreERT2 line they have generated and may constitute a useful new tool in the field. Although the results obtained provide first evidence for a role of RNase4 in nociceptors and suggest its absence contributes to faster recovery after peripheral injury, we feel the study at the present stage substantially lacks quantified analyses to support working models and findings. Several claims lack unequivocal demonstration, particularly with regards to segregation between RNase4 autonomous and non cell-autonomous functions.

Major concerns and specific points that could be addressed to strengthen the study are as follow:

Figure 1. In panel d-e, RNAscope is performed to confirm that RNase4 expression is selective of nociceptors. However, the quantification is performed by defining nociceptors solely based on their smaller soma size. C-LTMR are also of small diameter, which could bias the quantification. Therefore, further analysis is needed using bona fide markers of nociceptors, C-LTMR or myelinated LTMR subtypes (TrkA, CGRP, SST, TH, NF200, ...). These, with the already performed costaining with PV, should also be quantified to allow the reader to evaluate the specificity of RNase4 expression in situ.

Panel b suggests that RNase4 is found in a small fraction of the entire nociceptive population. Whether this observation can be replicated in situ is also important to demonstrate before diving into subsequent results.

Panel f shows that RNase4 also exists in human. The meaning of the indicated values in the graph is unclear.

Figure 2. Panel b showing expression of RNase4 in section of adult mice appears redundant with the image shown in panel c.

Panel e shows that many genes are misregulated in adult cKO-BR mice. Among the DEG identified, the authors cite Cpeb3 as a downregulated target whose deficiency is described in the literature to increase TrpV1 expression, however TrpV1 is not found among DEG in this study. Functional components of the nociceptors such as potassium voltage-gated channels are also misregulated, which is likely to impact the electrophysiological properties of nociceptors, but the behaviour tests in panel g show that cKO-BR mice behave as controls. Could it indicate that the cut-off chosen to define DEG was too loose? The authors could examine by IF/ISH the expression of some of the identified DEG in cKO-BR mice to validate these scRNAseq data.

In panel f, genes such as Kcnf1 or Hcn3 appear deregulated in cLTMR neurons, and Kcng3 in PEP2 nociceptors. However, RNase4 appear not expressed in these cell types (Fig.1c and 5a). This should be mentioned here as RNase4 is reported to act also as a secreted molecule and may thus have cell-autonomous and non-cell autonomous effects, and as the authors suggest it is the case during nerve injury.

In panel g, the authors show that mechanical and thermal sensitivity is unaltered in cKO-BR mice. The significance of this observation is hard to evaluate as RNase4 expression during development has not been approached.

In the very last sentence of the introduction (line 71), the authors state that the identity of nociceptors is affected. While transcriptional alterations are observed in cKO-BR, whether they are linked to a switched or aberrant identity is not demonstrated.

Figure 3. The Prdm12CreERT2 line is presented as a line useful to conditionally selectively delete RNase4 in nociceptors at adulthood. However, as previously reported by the authors, scRNA-seq data in panel 3c show that Prdm12 is also expressed in C-LTMR. Co-staining of the tdTomato-reporter with a marker of C-LTMR should thus be performed and quantified to appreciate the degree of overlap within the C-LTMR neurons at adulthood. Similarly, as the scRNA-seq data reveal Prdm12 expression in NF1, co-staining with a NF marker would be useful to validate this unexpected observation.

In panel d, a Prdm12CreERT2;Ai14 E15.5 embryo whole-mount immunostained for tdTomato and TrkA is shown. As this procedure does not allow to fully appreciate the degree and specificity of the overlap between tdTomato and TrkA, a quantified analysis of this overlap on DRG cross-sections could be shown in addition to fully validate this Prdm12CreERT2 line.

In panels 3e and 3f, the authors show that Tomato staining is found in the spinal cord dorsal horn which receives nociceptive and mechanoreceptive inputs in dedicated lamina regions. To demonstrate that the Tomato-stained fibres selectively terminate in the dorsal horn region receiving nociceptive afferents, co-staining with appropriate lamina marker (PKC γ , IB4, ...) should be performed.

In panel h, RNAscope images of DRG sections of adult cKO-PR mice are shown. Given the low level of recombination observed following tamoxifen injection (42.5%, panel g), the decrease of RNase4 staining observed in this cKO-PR line following tamoxifen injection should be quantified.

In panel i, the authors assessed mechanical sensitivity using the Von Frey test, and thermal pain using the Hargreaves test. Their results show that cKO-PR mice exhibit a transient increase sensitivity to mechanical stimuli 7d days post tamoxifen injection. Why has the Hargreaves test, which is not described in the text, been performed at 30d post-tamoxifen injection

and not 7d?

Figure 4. In panel a-c, using the iPain database they have established previously, the authors show that RNase4 expression increases during the first week post-injury in the nerve crush injury model. Such finding would be stronger if experimentally validated.

In panels e-j, the authors perform Von Frey test and Hargreaves test following nerve crush injury on cKO-BR and cKO-PR mice. They show that both cKO and control mice exhibit as anticipated a reduction in sensory function following injury and that the absence of RNase4 accelerates the recovery in sensory function compared to controls. The results of the Hargreaves test show opposite trends at 14 days after injury. The authors should describe and comment this result in the text.

Figure 5. ScRNA-seq data in panel a show that in trigeminal neurons RNase4 is excluded from myelinated PEP2 nociceptive neurons. ScRNA-seq data from DRG neurons shown in Fig.1a also suggest this pattern of expression. These data would be stronger if validated in situ.

When characterizing by EM sciatic nerve features of control and cKO-PR mice 14 days after injury, the authors show in panel e that the axon diameter of neurons are smaller in the ipsilateral side, potentially due to the inclusion of newly formed, thinner axons. However, they also observe reduced axon diameter in the contralateral side, which does not allow this potential explanation to stand.

In panel h,i, the authors show that there is an increased presence of misfolded myelin structures in a subset (5%) of myelinated fibers in the cKO-PR mice compared to controls. Is there a selective type of myelinated fiber affected? Is it possible that these affected myelinated fibers represent A δ -nociceptive fibers?

Panel p shows a model of RNase4 mode of action in nerve homeostasis and myelination, showing that it is released by the nerves to regulate nerve homeostasis and myelination. Is there any clue that RNase4 is indeed found in the sciatic nerve environment or at least secreted by somatosensory neurons?

Discussion

Line 266. The authors state that RNase4 plays a key role in maintaining the normal conductivity of unmyelinated nociceptors, however no experiment is performed to support this conclusion.

The discussion section extensively develops regarding the role of RNase4 in angiogenesis and its potential contribution to the nerve regeneration phenotype observed in cKO-PR mice. Could angiogenesis in the crush model in cKO-PR mice be quantified using EM images previously taken for analysis in figure 5?

Minor points

- Typo line 98 "Fig.1g" "Fig.1f"
- Typo line 103 "containg" "containing"
- Typo figure 5k, remark should be replaced by remark.
- In the methods section, lines 498 to 501 indicate that RNase4 is significantly decreased following TAM injection in cKO-PR mice, however no quantitative and statistical methods are shown to assess this reduction properly.
- The sentence at lines 509 to 511 should be rephrased.

Reviewer #3

(Remarks to the Author)

Reviewer #4

(Remarks to the Author)

This study employs a multifaceted approach combining genetic knockout models, single-cell transcriptomic profiling, pain behavioral assays, and electron microscopy. These investigations culminate in the identification of RNase4 as a previously unrecognized regulator of nociceptor biology, uncovering its dual roles: (1) a cell-autonomous function in maintaining

nociceptor function, and (2) non-cell-autonomous effects on the structural integrity of adjacent myelinated axons. While these findings provided a novel regulator of nociceptor biology, critical mechanistic insights and methodological and descriptive rigor are required to support and strengthen the broad claims.

Major concerns and recommendations

1. Data-interpretation discrepancy

There is a fundamental contradiction between the textual description and graphical presentation of mechanical hypersensitivity in animal model of nerve crush injury. In lines 208-215, the manuscript states that "Following nerve injury, cKO-BR and cKO-PR mice, along with their control littermates, exhibited the anticipated moderate and transient reduction in sensory function in response to mechanical stimuli." but Figure 4f and 4i demonstrate increased mechanical withdrawal thresholds (indicating hyposensitivity) in 7 days after nerve crush injury model compared to control mice. In lines 232-233, "However, the regenerated axon diameters were smaller (Fig. 5f), potentially due to the inclusion of newly formed, thinner axons", but Fig. 5f demonstrates the number of axons. This inconsistency directly undermines the central claim regarding RNase4's role in pain modulation.

2. Insufficient mechanism elucidation

The molecular mechanism by which RNase4, as an RNase, regulates nociceptor function remains unclear (e.g., target RNAs, downstream signaling). Although the authors performed scRNA-seq to find the role of RNase4 on the transcriptional phenotype of the nociceptor, they did not find the key downstream signaling due to they chose the wrong transgenic mice (Baf53bCre;Rnase4fl/fl (cKO-BR)). They should choose (Prdm12CreERT2;Rnase4fl/fl (cKO-PR)), which displays a transiently increases mechanical pain sensitivity. The authors provided a suppose that RNase5 serves as a compensatory factor for RNase4 in the discussion section. However, they do not identify the upregulation of RNase5 in cKO-BR mice. It is not even clear if RNase5 is expressed in nociceptors.

Recommendations:

- 1) Perform scRNA-seq in the nociceptor of cKO-PR mice after TAM injection for 7 days.
- 2) Assess RNA stability or protein levels of key pain-related genes (e.g., TRPV1, Nav1.8) in RNase4-deficient mice (cKO-PR) to link RNase4 activity to nociceptor function.
- 3) They should explore the expression of RNase5 in their scRNA-seq dataset and iPain single-cell transcriptomic atlas.

3. Inadequate statistical power in electron microscopic analysis

The ultrastructural analysis of myelinated axons in Figure 5 is based on data from only two cKO-PR mice (n = 2 biological replicates). This limited sample size precludes meaningful statistical inference, and renders the claimed "altered myelinated axon organization" biologically unreliable.

Minor issues

1. The author should explore the co-expression of Rnase4 and S100b or NF200, the broad marker genes of NFs because Pvalb just expressed in a kind of NFs.
2. In line 103, "Short Conditional intrON (SCON) cassette20 containg LoxP recombination sites" containg should be "containing"

Version 1:

Reviewer comments:

Reviewer #1

(Remarks to the Author)

The authors have adequately addressed my concerns and made appropriate revisions to strengthen the manuscript. I have no further questions.

(Remarks on code availability)

Reviewer #2

(Remarks to the Author)

We acknowledge that the authors have effectively addressed most of our previous concerns regarding their study on the role of RNase4 in nociceptor homeostasis. The inclusion of novel snRNA-seq data from the cKO-PR mice, together with the evidence indicating that RNase4 may exert its effects through modulation of the PI3K-AKT pathway, substantially strengthens the manuscript. Overall, these additions significantly enhance the impact of the study, and we therefore support its publication.

Minor points to be fixed:

- Fig 1, panels 1g, h and i are not referenced in the text.
Fig 1, panel j is not described and not referenced in the text.
Line 168: Fig. 3h should be 3g.

Line 197. The reasoning behind testing the effect of RNase4 effects on the PI3K–AKT pathway should be better introduced this should be explained.

Fig3, panel F. The legend of the figure indicates that this is an RNase4 staining. This is not visible in this panel. Fig.7C is not referenced in the text.

Line 313. There is no panel “q” in Fig. 7.

(Remarks on code availability)

Reviewer #3

(Remarks to the Author)

(Remarks on code availability)

Reviewer #4

(Remarks to the Author)

The authors have provided clear and detailed responses to each of my concerns, and they have made appropriate adjustments to the manuscript. These modifications have improved the work's impact and clarity while thoroughly and persuasively addressing my earlier criticisms. The revisions they have made significantly improve the manuscript. I have no more comments on this work, and I believe it is now well-suited for publication in Nature Communications.

(Remarks on code availability)

The code provide a README file. And i could install the dependencies of this code. However, due to the absence of the original files required to run the code, it is not possible to confirm whether the script can be reproduced.

REVIEWER COMMENTS

We thank the reviewers for their constructive feedback, which has helped improve the manuscript. Detailed responses to each comment are provided below in blue, with references to the corresponding revised sections of the text.

Reviewer #1 (Remarks to the Author):

This manuscript identifies RNase4 as selectively expressed in unmyelinated nociceptors and reveals its dual role in nociceptor function and nerve homeostasis. Through an integrated approach combining single-cell transcriptomics, genetic mouse models, and functional assays, the authors demonstrate that RNase4 regulates both nociceptor molecular identity and influences neighboring myelinated axons. While conditional deletion of Rnase4 during development alters gene expression profiles without affecting baseline pain sensitivity, adult-onset deletion causes transient mechanical hypersensitivity. Following nerve injury, Rnase4 deletion accelerates functional recovery and remyelination. The findings establish RNase4 as an important regulator of nociceptor biology with both cell-autonomous and non-cell autonomous effects. However, the molecular mechanisms underlying these effects remain inadequately explored.

Specific comments:

1. The authors should perform co-localization studies using established nociceptor markers with RNase4 RNA FISH. Additionally, protein-level detection of RNase4 would strengthen the findings, as protein is the functional entity.

We agree with the reviewer that further characterization of RNase cell type–restricted expression can be more thoroughly explored using mRNA hybridization techniques. To this end, we employed RNAscope, a highly sensitive and specific in situ hybridization (ISH) assay that allows spatial detection of target RNA molecules with single-cell resolution RNAScope.

The following text has been added to the manuscript from line 90 to 100.

“To further characterize this cell type–restricted expression, we performed RNAscope on cross-sections of dorsal root ganglia (DRG) from adult animals (Fig. 1). Co-labeling used established markers: Calca (peptidergic nociceptors, PEPs), SST (somatostatin-positive neurons), TH

(cLTMRs), NF200/NF-H and PV (myelinated neurofilament neurons, NFs), peripherin (Prph; small-diameter, unmyelinated neurons), and Prdm12 (nociceptor lineage marker).

RNase4 showed complete overlap with SST+ neurons and partial colocalization (~40-50%) with Calca+ peptidergic neurons, while TH+ cLTMRs, which share a developmental nociceptive lineage but mediate pleasant touch, were entirely RNase4-negative (Fig. 1d-f). Minimal overlap occurred with NF200+ populations and none with PV+ myelinated neurons, with RNase4 enriched in Prdm12+/peripherin+ small-diameter neurons. These findings establish RNase4 as a selective marker of specific unmyelinated nociceptor subtypes.”

2. In Fig 1g, comparative expression data showing differential expression between neuronal subtypes or tissues would provide better context for RNase4 specificity.

In the previous version of our manuscript, the human DRG dataset was sufficient for bulk-level analysis but did not allow for deconvolution into single cells into cell types. However, the recently published single-nucleus DRG atlas (Bhuiyan et al., bioRxiv 2025, Figure 1) does capture RNASE4 expression in human DRG neurons, confirming its presence and distribution among cell-types. Those data recapitulate the mouse RNase4 expression pattern we have characterized.

The following text has been added to the manuscript from line 101 to 104:

“Bulk RNA-seq analysis of human dorsal root ganglia (DRG)¹⁹ confirmed RNase4 expression in human tissue, with a selective nociceptor-enriched pattern evident in recent single-cell atlas of human somatosensory neurons²⁰ (Fig. 1k,l). These findings demonstrate evolutionary conservation of RNase4 expression across species, suggesting an important role in nociceptor biology.”

For tissue-level expression, the Human Protein Atlas also reports RNASE4 diverse tissues RNASE4 protein expression summary - The Human Protein Atlas.

3. Fig 2b and 2c rely on Nissl staining, which labels all neurons. Using DRG neuron-specific markers would provide better characterization of the cells expressing RNase4.

This comment has now been addressed and the results are presented in Figure 1.

4. The authors highlight differentially expressed genes in Figures 2e-f but fail to present comprehensive functional enrichment analysis of these genes. The manuscript mentions alterations in zinc finger proteins, transcription factors, and immune response pathways, but these findings are not adequately presented in figures or supplementary data.

The authors apologize for the omission and now present in the supplementary file an enrichment analysis.

5. In Fig 2g, the nature of individual data points is unclear, and there appears to be inconsistency in the number of data points between left and right panels.

We are grateful to the reviewer for noticing this, this has now been corrected.

6. The authors should examine whether RNase4 expression changes during the course of behavioral testing (von Frey and Hargreaves tests), which would help connect molecular changes with behavioral outcomes.

The authors have addressed this point from line 227 to 241 (Fig. 6a-c). Using the iPain atlas, we report RNASE4 expression dynamics after nerve injury across 6 timepoints up to 28 days post-injury. In parallel, longitudinal behavioral assessments (von Frey for mechanical sensitivity, Hargreaves for thermal sensitivity) at 0, 7, 14, and 30 days pre- and post-RNASE4 ablation (Fig. 3h) directly link the changes in RNASE4 expression with alterations in sensory thresholds.

7. Regarding the observed thicker myelin and smaller axon diameter in contralateral nerves of cKO-PR mice (Fig. 5c-f), it is important to clarify whether these are developmental effects or consequences of adult Rnase4 deletion. If the latter, the time course of these structural changes following deletion should be determined.

For the cKO-PR mice, RNASE4 deletion is temporally controlled and was induced in adulthood. In this study, recombination was performed in adult mice prior to injury, and the nerves were examined two weeks post-injury. Therefore, the structural changes observed in both ipsilateral and contralateral nerves arise from acute adult deletion rather than developmental effects.

Longer or shorter time-course experiments were not feasible due to animal-number restrictions, but the current design isolates the effect of adult RNASE4 loss.

8. The authors propose a compensatory relationship between RNase4 and RNase5 (angiogenin) in nerve regeneration but provide no experimental evidence. Examining RNase5 expression in Rnase4 knockout conditions would substantiate this hypothesis.

We thank the reviewer for this comment. We performed the suggested experiment and did not detect RNase5 expression under RNase4 knockout conditions (Supplementary Fig. 6). In line with this finding, we have corrected the statement suggesting compensation by RNase5.

9. As RNase4 is described as a secreted protein, rescue experiments administering recombinant RNase4 would demonstrate whether it has direct protective effects and strengthen the causal relationship between RNase4 and the observed phenotypes.

We thank the reviewer for this insightful suggestion. Rescue experiments using recombinant RNase4 would indeed provide an important test of causality. However, due to limited animal availability and the need for careful optimization of dosing and delivery, these experiments could not be included in the present study. We agree that this represents an important direction for future investigation and plan to pursue this approach in future work.

10. The study lacks mechanistic explanation: Does RNase4 function through its enzymatic activity or through other molecular pathways? Is it modulating known signaling pathways such as AXL?

We thank the reviewer for this important point. We have now investigated PI3K, AKT, mTOR and AXL in cell culture, treated with either inhibitor of the signaling pathways with and without Rnase4.

The following text has been added to the manuscript from line 192 to 225:

“RNase4 cell non-autonomous action: modulation of PI3K–AKT–mTOR signaling

To determine how RNase4 (R4), when present in the extracellular environment, influences intracellular signaling in surrounding cells, we treated ND7/23 cells with RNase4 alone or in combination with inhibitors of the PI3K/Akt/mTOR pathway. We assessed phosphorylation of Akt (Ser473) and mTOR (Ser2448) as readouts of pathway activity²⁸.

In the first experimental dataset (Fig. 5a), inhibition of Akt (MK-2206) and PI3K (LY294002) markedly reduced p-Akt levels compared with vehicle control, confirming effective pathway suppression. When ND7/23 cells were co-treated with RNase4, p-Akt levels remained similarly reduced in all inhibitor conditions, indicating that RNase4 does not override pharmacological pathway blockade. Notably, RNase4 decreased p-Akt/Akt ratios relative to DMSO alone, suggesting that extracellular RNase4 can dampen basal Akt activation. This reduction corresponds to an estimated 60% (Fig. 5a,b) ($P=0.0072$; $n=3$ biological replicates) decrease in the p-Akt/Akt ratio, indicating a significant inhibitory effect of RNase4. Interestingly, while LY294002 robustly and significantly suppressed p-Akt levels, wortmannin led to a decrease of p-Akt level though it was not significant, and lead to a moderate suppression of around 65% (not significant) when associated with Rnase4 (Fig. 5b), despite targeting the same pathway. Furthermore, no significant effect of inhibitor rapamycin alone was found at any concentration though surprisingly the Akt levels increased by 40% in line with rapamycin-mediated inhibition of mTORC1 can activate PI3K/Akt signaling by relieving a negative feedback loop²⁹, which

may explain the upward trend in Akt levels observed in the same experiment. The addition of RNase4 to rapamycin treated cells diminished p-Akt levels to an estimate of 60% compared to rapamycin only (P=0.011 for 5nM condition and P=0.018 for 10nM condition, Fig. 5b). These data indicate that RNase4 exerts a consistent Akt-suppressive effect on ND7/23 cells. Investigating the effect of RNase4 on mTOR phosphorylation, particularly at Ser2448 (Fig. 5c-d), we found a subtle decrease of around 20%, though not significant. Expectedly, mTOR level was found to be significantly reduced (P= 0.0082) by the PI3K/Akt pathway inhibitor LY294002 (Fig. 5c-d). β III-tubulin levels were comparable across all conditions (Fig. 5e-f), verifying equal protein loading and supporting the reliability of the phosphorylation analyses.

Notably, while Akt and mTOR expression were detectable, Axl receptor protein, a known PI3K–Akt–mTOR modulator, was undetectable in both ND7/23 cells and primary sensory neuron cultures (Supplementary Fig. 4), indicating that the effects of RNase4 on this pathway in the DRG-derived context are Axl-independent. Taken together, our results demonstrate that RNase4 may also exert its effect through modulating the PI3K–AKT–mTOR pathway in the DRG, in line with its previously reported effects in cancer cells³⁰, but its effects here may not be through the Axl receptor.”

We hope that the reviewer finds the revised manuscript satisfactory and suitable for publication in *Nature Communications*.

Reviewer #2 (Remarks to the Author):

RNase4 is known to be involved in a broad range of biological processes, including the regulation of intra- or extra-cellular RNA metabolism as well as antiviral, antibacterial, and antifungal activities, neurogenesis, angiogenesis, immune modulation and host defence, is selectively expressed in neurons of the nociceptive lineage. In this manuscript, the authors report that RNase4 is selectively expressed in sensory ganglia in the unmyelinated nociceptive lineage. They analysed the consequences of its loss of function using both a conditional KO and an inducible cKO strategy with a novel Prdm12CreERT2 line they have generated and may constitute a useful new tool in the field. Although the results obtained provide first evidence for a role of RNase4 in nociceptors and suggest its absence contributes to faster recovery after peripheral injury, we feel the study at the present stage substantially lacks quantified analyses to support working models and findings. Several claims lack unequivocal demonstration, particularly with regards to segregation between RNase4 autonomous and non cell-autonomous functions.

Major concerns and specific points that could be addressed to strengthen the study are as follow:

Figure 1. In panel d-e, RNAscope is performed to confirm that RNase4 expression is selective of nociceptors. However, the quantification is performed by defining nociceptors solely based on their smaller soma size. C-LTMR are also of small diameter, which could bias the quantification. Therefore, further analysis is needed using bona fide markers of nociceptors, C-LTMR or myelinated LTMR subtypes (TrkA, CGRP, SST, TH, NF200, ...). These, with the already performed costaining with PV, should also be quantified to allow the reader to evaluate the specificity of Rnase4 expression in situ.

Panel b suggests that RNase4 is found in a small fraction of the entire nociceptive population. Whether this observation can be replicated in situ is also important to demonstrate before diving into subsequent results.

We appreciate the reviewer's critical evaluation and agree that defining nociceptor populations solely based on smaller soma size can introduce quantification bias, as C-LTMRs are also of small diameter. To strengthen the study, we have performed further RNAscope analysis using bona fide molecular markers for distinct DRG neuron subtypes.

The following text has been added to the manuscript from line 90 to 100:

“To further characterize this cell type–restricted expression, we performed RNAscope on cross-sections of dorsal root ganglia (DRG) from adult animals (Fig. 1). Co-labeling used established markers: Calca (peptidergic nociceptors, PEPs), SST (somatostatin-positive neurons), TH (cLTMRs), NF200/NF-H and PV (myelinated neurofilament neurons, NFs), peripherin (Prph; small-diameter, unmyelinated neurons), and Prdm12 (nociceptor lineage marker).

RNase4 showed complete overlap with SST+ neurons and partial colocalization (~40-50%) with Calca+ peptidergic neurons, while TH+ cLTMRs, which share a developmental nociceptive lineage but mediate pleasant touch, were entirely RNase4-negative (Fig. 1d-f). Minimal overlap occurred with NF200+ populations and none with PV+ myelinated neurons, with RNase4 enriched in Prdm12+/peripherin+ small-diameter neurons. These findings establish RNase4 as a selective marker of specific unmyelinated nociceptor subtypes.”

Panel f shows that RNase4 also exists in human. The meaning of the indicated values in the graph is unclear.

This has now been addressed (Figure 1k in the updated manuscript)

Figure 2. Panel b showing expression of RNase4 in section of adult mice appears redundant with the image shown in panel c.

Panel has been removed.

Panel e shows that many genes are misregulated in adult cKO-BR mice. Among the DEG identified, the authors cite Cpeb3 as a downregulated target whose deficiency is described in the literature to increase TrpV1 expression, however TrpV1 is not found among DEG in this study. Functional components of the nociceptors such as potassium voltage-gated channels are also misregulated, which is likely to impact the electrophysiological properties of nociceptors, but the behaviour tests in panel g show that cKO-BR mice behave as controls. Could it indicate that the cut-off chosen to define DEG was too loose? The authors could examine by IF/ISH the expression of some of the identified DEG in cKO-BR mice to validate these scRNAseq data.

The changes identified in the single-cell data were confirmed using RNAscope for the potassium ion channels Kcnf1 and Kcng3 (Figure 2f in the updated manuscript). We agree with the reviewer that the behavioral results do not necessarily reflect the transcriptional changes.

We also added the corresponding text 130 to 145:

“The analysis also highlighted notable changes in cytoskeleton-related proteins and ribosomal components (Supplementary Fig. 1a,b). Alterations were observed in microtubule-binding motor protein *Dync2h1*, *Mapt* (microtubule-associated protein tau), and tubulin variant *Tubd1*. Furthermore, eleven ribosomal proteins were affected, including *Mrpl4*, *Rpl9*, *Rps10*, *Rps27*, and *Mrps1*. Significant changes were also noted in translation-related factors such as *Eif2s1* (a translation initiation factor), *Gspt1* (a translation termination factor), and *Cpeb3*, a negative regulator of translation with known function on *Trpv1* expression²⁴. These findings indicate widespread modifications in cellular architecture and protein synthesis machinery as a response to conditional mutation. Finally, focusing on ion channels within nociceptors, we identified differences in expression levels of potassium voltage-gated ion channels *Kcnf1* (Kv5.1) and *Kcng3* (Kv6.3) (Fig. 2e, f). These channels are integral to the potassium channel machinery that regulates membrane potential and facilitates the return to a negative resting potential following action potential. Their reduction in RNase4 cKO neurons may affect Kv channel composition and potentially influence neuronal excitability^{25,26}. Interestingly, RNase4 deletion altered gene expression in cLTMRs despite the absence of RNase4 in this population, an observation that could be consistent with paracrine effects of RNase4 on neighboring cells.”

In panel f, genes such as *Kcnf1* or *Hcn3* appear deregulated in cLTMR neurons, and *Kcng3* in PEP2 nociceptors. However, RNase4 appear not expressed in these cell types (Fig.1c and 5a). This should be mentioned here as RNase4 is reported to act also as a secreted molecule and may thus have cell-autonomous and non-cell autonomous effects, and as the authors suggest it is the case during nerve injury.

We thank the reviewer for this constructive comment and have now introduced the secreted nature of RNase4 earlier in the text from line 143 to 145.

“Interestingly, RNase4 deletion altered gene expression in cLTMRs despite the absence of RNase4 in this population, an observation that could be consistent with paracrine effects of RNase4 on neighboring cells.”

In panel g, the authors show that mechanical and thermal sensitivity is unaltered in cKO-BR mice. The significance of this observation is hard to evaluate as RNase4 expression during development has not been approached.

We thank the reviewer for this constructive comment. We had attached the expression profile of Rnase4 during development as shown in the plots below:

Panel legend: The expression dynamics of Rnase4 and Baf (Actl6b) along the embryonic days extracted from Faure et al., 2020.

Panel legend: The expression dynamics of Rnase4 and Baf (Actl6b) along the embryonic days extracted from Sharma et al., 2020.

In the very last sentence of the introduction (line 71), the authors state that the identity of nociceptors is affected. While transcriptional alterations are observed in cKO-BR, whether they are linked to a switched or aberrant identity is not demonstrated.

We thank the reviewer for pointing this out and have removed the sentence to avoid overinterpretation.

Figure 3. The Prdm12CreERT2 line is presented as a line useful to conditionally selectively delete RNase4 in nociceptors at adulthood. However, as previously reported by the authors, scRNA-seq data in panel 3c show that Prdm12 is also expressed in C-LTMR. Co-staining of the tdTomato-reporter with a marker of C-LTMR should thus be performed and quantified to appreciate the degree of overlap within the C-LTMR neurons at adulthood. Similarly, as the scRNA-seq data reveal Prdm12 expression in NF1, co-staining with a NF marker would be useful to validate this unexpected observation.

We now provide data showing co-localization of RFP with TH (C-LTMRs) and absence of co-localization with PV (Supplementary Figure 2 in the updated manuscript).

In panel d, a Prdm12CreERT2;Ai14 E15,5 embryo whole-mount immunostained for tdTomato and TrkA is shown. As this procedure does not allow to fully appreciate the degree and specificity of the overlap between tdTomato and TrkA, a quantified analysis of this overlap on DRG cross-sections could be shown in addition to fully validate this Prdm12CreERT2 line.

We now show that TrkA co-localizes with RFP in DRG cross-sections, confirming the overlap (Fig. 3d).

In panels 3e and 3f, the authors show that Tomato staining is found in the spinal cord dorsal horn which receives nociceptive and mechanoreceptive inputs in dedicated lamina regions. To demonstrate that the Tomato-stained fibers selectively terminate in the dorsal horn region receiving nociceptive afferents, co-staining with appropriate lamina marker (PKC γ , IB4,...) should be performed.

We agree that defining the laminar termination is important. We have now performed co-staining of RFP fibers with IB4 and CGRP (Fig. 3e).

The corresponding text was added in the manuscript from line 161 to 164.

“RFP+ cells and fibers show clear exclusion with PV (Fig. 3d) and co-localization with TrkA (Fig. 3d), IB4 and CGRP (Fig. 3e), indicating that they terminate within laminae associated

with nociceptive afferents. This confirms the selective projection of Prdm12-lineage neurons to I-II regions of the dorsal horn.”

In panel h, RNAscope images of DRG sections of adult cKO-PR mice are shown. Given the low level of recombination observed following tamoxifen injection (42.5%, panel g), the decrease of RNase4 staining observed in this cKO-PR line following tamoxifen injection should be quantified.

This has now been quantified (Fig. 3g). Number of RNase4 / Nissl neurons were around 55% in control and 19% in cKO-PR as quantified from $n = 3$ mice per group, significance tests showed $p < 0.001$.

In panel i, the authors assessed mechanical sensitivity using the Von Frey test, and thermal pain using the Hargreaves test. Their results show that cKO-PR mice exhibit a transient increase sensitivity to mechanical stimuli 7d days post tamoxifen injection. Why has the Hargreaves test, which is not described in the text, been performed at 30d post-tamoxifen injection and not 7d?

The Hargreaves test was performed at the same timepoints as the von Frey tests, including 7 days post-tamoxifen, but did not show significant differences between genotypes at that timepoint. Therefore, only the 30-day data are presented.

A sentence clarifying this point has now been added to the manuscript from line 173 to 177:

“Thermal sensitivity, assessed with the Hargreaves test, showed no genotype-dependent differences at the intermediate time points; therefore, only the 30-day Hargreaves data are presented. These findings suggest RNase4 contributes to adult nociceptor function, although its absence causes only transient effects (likely due to compensatory mechanisms). The data also hint at a role for RNase4 in nociceptor homeostasis.”

Figure 4. In panel a-c, using the iPain database they have established previously, the authors show that RNase4 expression increases during the first week post-injury in the nerve crush injury model. Such finding would be stronger if experimentally validated.

We understand the reviewer’s advice; however, the iPain resource is based on previously published single-cell transcriptomic datasets generated from injured tissue, not on predictive models. These datasets originate from multiple laboratories and injury models and therefore already represent experimentally derived measurements. We developed iPain in part to reduce

duplication of animal experiments in pain research, in line with the 3R principles, and we hope the reviewer supports this approach.

In panels e-j, the authors perform Von Frey test and Hargreaves test following nerve crush injury on cKO-BR and cKO-PR mice. They show that both cKO and control mice exhibit as anticipated a reduction in sensory function following injury and that the absence of RNase4 accelerates the recovery in sensory function compared to controls. The results of the Hargreaves test show opposite trends at 14 days after injury. The authors should describe and comment this result in the text.

We thank the reviewer for the comments.

The following text has been added to the manuscript from line 265 to 275:

“Von Frey testing revealed that both cKO-BR, cKO-PR (Fig. 6f,i), and control mice displayed the expected reduction in mechanical sensitivity following sciatic nerve crush, followed by a progressive recovery over time. In both knockout lines, recovery of mechanical thresholds occurred faster than in littermate controls, indicating that loss of RNase4 facilitates restoration of tactile sensitivity after injury. In contrast, Hargreaves testing of thermal sensitivity showed an opposite trend between genotypes cKOs at 14 days post-injury, with cKO-BR mice displaying a delay in thermal recovery compared to controls while cKO-PR fully recovered. This divergence suggests that RNase4 modulates regeneration of mechanical and heat pathways in a differential, modality-specific manner, and highlights that accelerated recovery of tactile function in the knockouts does not necessarily generalize to thermal nociception at all stages of the regeneration process.”

Figure 5. ScRNA-seq data in panel a show that in trigeminal neurons RNase4 is excluded from myelinated PEP2 nociceptive neurons. ScRNA-seq data from DRG neurons shown in Fig.1a also suggest this pattern of expression. These data would be stronger if validated in situ. Although we agree that in-situ validation of PEP2 neurons would be informative, this analysis is technically challenging because no single reliable marker labels the entire PEP2 population. Available markers such as Penk and Trpm8 identify only subsets of these neurons to our knowledge, which would limit the interpretability of such an experiment. Importantly, the exclusion pattern of RNase4 is supported by two independent scRNA-seq datasets from DRG and trigeminal ganglia, which already provides strong convergent evidence. We hope the reviewer finds this clarification satisfactory.

When characterizing by EM sciatic nerve features of control and cKO-PR mice 14 days after injury, the authors show in panel e that the axons diameter of neurons are smaller in the ipsilateral side, potentially due to the inclusion of newly formed, thinner axons. However, they also observe reduced axon diameter in the contralateral side, which does not allow this potential explanation to stand.

We characterized both ipsilateral and contralateral nerves to distinguish between effects related to injury and those related to the conditional deletion. The reduction in axon diameter observed on the ipsilateral side may reflect the presence of newly formed axons after injury. However, the reduced axon diameter on the contralateral side indicates that this phenotype is not solely injury-dependent and is instead attributed to the mutation itself.

In panel h, the authors show that there is an increased presence of misfolded myelin structures in a subset (5%) of myelinated fibers in the cKO-PR mice compared to controls. Is there a selective type of myelinated fiber affected? Is it possible that these affected myelinated fibers represent A δ -nociceptive fibers?

We thank the reviewer for this interesting point. We examined whether a specific class of myelinated fibers was selectively affected. While we cannot definitively assign subtype identity to the affected fibers, we analyzed the axons by diameter range. This showed that the increase in myelin misfolding is predominantly present in intermediate-diameter axons, in both genotypes. Therefore, the effect does not appear in uniform across all myelinated fibers. This could indeed be some A δ fibers. We have included this point in our updated manuscript (Figure 7i).

The following text has now been added to the manuscript from line 293 to 296:

“Notably, the incidence of aberrant myelin structures, including focal hypermyelination, myelin infolds, and degenerating myelin, was significantly elevated in cKO-Rnase4 nerves compared to controls (Fig. 7h,i). This increase in myelin misfolding was predominantly observed in intermediate-diameter fibers, with both genotypes showing similar distributions of myelin defects (Fig. 7i).”

Panel p shows a model of RNase4 mode of action in nerve homeostasis and myelination, showing that it is released by the nerves to regulate nerve homeostasis and myelination. Is there any clue that RNase4 is indeed found in the sciatic nerve environment or at least secreted by somatosensory neurons?

RNase4 is a secreted ribonuclease by structural domain, and multiple transcriptomic datasets show robust expression in sensory neurons. The functional phenotype in the sciatic nerve after RNase4 deletion further supports its presence and role locally. We also added a new figure (Fig.5) showing that RNase4 activates the AKT and mTOR pathways, further supporting its functional role in nerve homeostasis.

Accordingly, we added the following text line 307 to line 313:

“Importantly, western blot analysis revealed increased pAkt/Akt levels upon RNase4 inhibition, indicating enhanced activation of the PI3K–AKT pathway (Fig.5). Given the established role of AKT signaling in axonal growth and regeneration^{31,32} these data provide a mechanistic link between RNase4 loss and the observed acceleration of axonal regrowth. Together, our findings suggest that RNase4 normally restrains regenerative signaling, and that its absence promotes axonal regeneration, potentially through AKT-dependent pathways, while simultaneously perturbing Schwann cell homeostasis and myelin remodeling (Fig. 7q).”

Discussion

Line 266. The authors state that RNase4 plays a key role in maintaining the normal conductivity of unmyelinated nociceptors, however, no experiment is performed to support this conclusion.

We thank the reviewer for this comment. The sentence has now been reformulated to avoid overinterpretation and to reflect only the experimental observations.

The discussion section extensively develops regarding the role of RNase4 in angiogenesis and its potential contribution to the nerve regeneration phenotype observed in cKO-PR mice. Could angiogenesis in the crush model in cKO-PR mice be quantified using EM images previously taken for analysis in figure 5?

Unfortunately, the images obtained are not suitable for an angiogenesis analysis.

Minor points

-Typo line 98 “Fig.1g” to “Fig.1f” Those have been addressed

-Typo line 103 “containg” to ”containing” Those have been addressed

-Typo figure 5k, remark should be replaced by remak. Those have been addressed

-In the methods section, lines 498 to 501 indicate that RNase4 is significantly decreased following TAM injection in cKO-PR mice, however no quantitative and statistical methods are shown to assess this reduction properly. Those have been addressed

-The sentence at lines 509 to 511 should be rephrased. Those have been addressed.

We are grateful to the reviewer for their comments and hope that they are satisfied with the changes made to the manuscript.

Reviewer #3 (Remarks to the Author):

Reviewer #4 (Remarks to the Author):

This study employs a multifaceted approach combining genetic knockout models, single-cell transcriptomic profiling, pain behavioral assays, and electron microscopy. These investigations culminate in the identification of RNase4 as a previously unrecognized regulator of nociceptor biology, uncovering its dual roles: (1) a cell-autonomous function in maintaining nociceptor function, and (2) non-cell-autonomous effects on the structural integrity of adjacent myelinated axons. While these findings provided a novel regulator of nociceptor biology, critical mechanistic insights and methodological and descriptive rigor are required to support and strengthen the broad claims.

The authors appreciate and are thankful for the reviewer's efforts to improve the manuscript.

Major concerns and recommendations

1. Data-interpretation discrepancy

There is a fundamental contradiction between the textual description and graphical presentation of mechanical hypersensitivity in animal model of nerve crush injury. In lines 208-215, the

manuscript states that "Following nerve injury, cKO-BR and cKO-PR mice, along with their control littermates, exhibited the anticipated moderate and transient reduction in sensory function in response to mechanical stimuli.", but Figure 4f and 4i demonstrate increased mechanical withdrawal thresholds (indicating hyposensitivity) in 7 days after nerve crush injury model compared to control mice. In lines 232-233, "However, the regenerated axon diameters were smaller (Fig. 5f), potentially due to the inclusion of newly formed, thinner axons", but Fig. 5f demonstrates the number of axons. This inconsistency directly undermines the central claim regarding RNase4's role in pain modulation.

We believe there is a misunderstanding in the interpretation of the data. We agree that increased withdrawal thresholds indicate reduced sensory function (hyposensitivity), which is the expected and classical outcome of the nerve crush model. This is consistent with our results at 7 days post-injury.

Regarding the axon analysis, the reviewer is correct that the axon diameter was quantified at 14 days post-injury, not at 7 days. The reference in the previous version of the manuscript mistakenly pointed to Fig. 5f instead of Fig. 5e. We have now corrected the figure references and revised the text to be in accordance with our results.

2. Insufficient mechanism elucidation

The molecular mechanism by which RNase4, as an RNase, regulates nociceptor function remains unclear (e.g., target RNAs, downstream signaling). Although the authors performed scRNA-seq to find the role of RNase4 on the transcriptional phenotype of the nociceptor, they did not find the key downstream signaling due to they chose the wrong transgenic mice (Baf53bCre;Rnase4fl/fl (cKO-BR),). They should choose (Prdm12CreERT2;Rnase4fl/fl (cKO-PR)), which displays a transiently increase in mechanical pain sensitivity. The authors provided a suppose that RNase5 serves as a compensatory factor for RNase4 in the discussion section. However, they do not identify the upregulation of RNase5 in cKO-BR mice. It is not even clear if RNase5 is expressed in nociceptors.

Regarding the sequencing data, we did not choose the wrong mice. We performed sequencing for both transgenic lines, but the cKO-PR dataset failed and at that time we did not have sufficient animals to repeat the experiment. Since the original submission, we have re-established the line and obtained enough animals to repeat the sequencing. We have now

successfully generated the cKO-PR dataset at 7 days post-tamoxifen injection and include the analysis in a new Figure 4.

Regarding RNase5, new analysis and staining were performed and are now added to the text from line 366 to 373 and the in the supplementary Fig.6.

“Although no overt pain phenotype was observed under normal conditions in the absence of RNase4 when deleted during embryogenesis, a notable mechanical allodynia emerged several days after acute deletion of RNase4 in adult mice. This phenotype was transient, resolving within a month, suggesting that nociceptors activate compensatory mechanisms to restore homeostasis. RNase5 (angiogenin), a structurally related RNase A superfamily member with established roles in neuroprotection^{36,37}, was undetectable in sensory neurons (Supplementary Fig. 6) in basal condition or in cKO-Rnase4 mice, excluding simple RNase superfamily compensation. “

Recommendations:

1) Perform scRNA-seq in the nociceptor of cKO-PR mice after TAM injection for 7 days.

This experiment has been done, and we have now created a new figure with the analysis of the cKO PR neurons.

2) Assess RNA stability or protein levels of key pain-related genes (e.g., TRPV1, Nav1.8) in RNase4-deficient mice (cKO-PR) to link RNase4 activity to nociceptor function.

We have performed the RNAscope for TRPV1 and could not detect any obvious variation with the removal of RNase4 in the cKO-PR.

3) They should explore the expression of RNase5 in their scRNA-seq dataset and iPain single-cell transcriptomic atlas.

We have examined RNase5 expressions in the iPain datasets, in our single-cell RNA-seq data, and by RNAscope. RNase5 is not expressed in DRG neurons in control nor in CKO conditions.

We have added this information to the revised manuscript from line 369 to 373.

“This phenotype was transient, resolving within a month, suggesting that nociceptors activate compensatory mechanisms to restore homeostasis. RNase5 (angiogenin), a structurally related RNase A superfamily member with established roles in neuroprotection^{33,34}, was undetectable

in sensory neurons (Supplementary Fig. 6) in basal condition or in cKO-Rnase4 mice, excluding simple RNase superfamily compensation.”

3. Inadequate statistical power in electron microscopic analysis

The ultrastructural analysis of myelinated axons in Figure 5 is based on data from only two cKO-PR mice (n = 2 biological replicates). This limited sample size precludes meaningful statistical inference, and renders the claimed "altered myelinated axon organization" biologically unreliable.

The reviewer notes that the ultrastructural analysis in Figure 5 was based on two biological replicates. We agree that this limited the statistical power. We have now increased the number of animals per group (n=4 control and n=3 cKO-PR) and updated the figure and statistical analysis accordingly.

Minor issues

1. The author should explore the co-expression of Rnase4 and S100b or NF200, the broad marker genes of NFs because Pvalb just expressed in a kind of NFs.

We agree with the reviewer that further characterization of RNase cell type–restricted expression can be more thoroughly explored using mRNA hybridization techniques. To this end, we employed RNAscope, a highly sensitive and specific in situ hybridization (ISH) assay that allows spatial detection of target RNA molecules with single-cell resolution RNAScope.

The following text has been added to the manuscript from line 90 to 100.

“To further characterize this cell type–restricted expression, we performed RNAscope on cross-sections of dorsal root ganglia (DRG) from adult animals (Fig. 1). Co-labeling used established markers: Calca (peptidergic nociceptors, PEPs), SST (somatostatin-positive neurons), TH (cLTMRs), NF200/NF-H and PV (myelinated neurofilament neurons, NFs), peripherin (Prph; small-diameter, unmyelinated neurons), and Prdm12 (nociceptor lineage marker).

RNase4 showed complete overlap with SST+ neurons and partial colocalization (~40-50%) with Calca+ peptidergic neurons, while TH+ cLTMRs, which share a developmental nociceptive lineage but mediate pleasant touch, were entirely RNase4-negative (Fig. 1d-f). Minimal overlap occurred with NF200+ populations and none with PV+ myelinated neurons, with RNase4 enriched in Prdm12+/peripherin+ small-diameter neurons. These findings establish RNase4 as a selective marker of specific unmyelinated nociceptor subtypes.”

2. In line 103, “Short Conditional intrON (SCON) cassette²⁰ containing LoxP recombination sites” containing should be “containing”

This has been changed.

We hope that the reviewer finds the revised manuscript satisfactory and suitable for publication in *Nature Communications*.

Rebuttal Second revision: Response to reviewer comments,

We would like to sincerely thank all the reviewers for their careful evaluation of our manuscript, for their supportive assessment of the study, and for recognizing the interest and significance of the findings.

REVIEWERS' COMMENTS

Reviewer #1 (Remarks to the Author):

The authors have adequately addressed my concerns and made appropriate revisions to strengthen the manuscript. I have no further questions.

Thank you for your careful evaluation of our manuscript and for your constructive feedback throughout the review process. We sincerely appreciate your time and thoughtful assessment.

Reviewer #2 (Remarks to the Author):

We acknowledge that the authors have effectively addressed most of our previous concerns regarding their study on the role of RNase4 in nociceptor homeostasis. The inclusion of novel snRNA-seq data from the cKO-PR mice, together with the evidence indicating that RNase4 may exert its effects through modulation of the PI3K–AKT pathway, substantially strengthens the manuscript. Overall, these additions significantly enhance the impact of the study, and we therefore support its publication.

We sincerely thank the reviewer for the careful re-evaluation of our manuscript and for the thoughtful and constructive feedback throughout the review process.

We are grateful that the reviewer recognizes the value of the newly included snRNA-seq data from the cKO-PR mice and the mechanistic insights linking RNase4 to modulation of the PI3K–AKT pathway. We agree that these additions significantly strengthen the study and clarify the role of RNase4 in nociceptor homeostasis.

We appreciate the reviewer's support for publication and the engagement in improving the impact of this work.

Minor points to be fixed:

1. Fig. 1, panels g, h, and i are not referenced in the text.
These panels are now explicitly referenced and described in the Results section.
2. Fig. 1, panel j is not described and not referenced in the text.
Panel j has now been described and appropriately referenced in the Results section.
3. Line 168: Fig. 3h should be Fig. 3g.
This typographical error has been corrected.
4. Line 197: The reasoning behind testing the effect of RNase4 on the PI3K–Akt pathway should be better introduced this should be explained.
We have revised the text to explicitly introduce the rationale for examining the PI3K–Akt–mTOR pathway, clarifying why this signaling axis was tested in the context of extracellular RNase4 activity.

5. Fig3, panel F. The legend of the figure indicates that this is an RNase4 staining. This is not visible in this panel.
The figure legend has been corrected to accurately reflect the content of the panel.
6. Fig. 7C is not referenced in the text.
Fig. 7C is now explicitly referenced and discussed in the Results section.
7. Line 313: There is no panel “q” in Fig. 7.
This has been corrected, and the reference now corresponds to the correct panel.

Reviewer #3 (Remarks to the Author):

Thank you for this information. We appreciate the transparency regarding the co-review process and fully support initiatives that promote training and recognition of Early Career Researchers in peer review.

Reviewer #4 (Remarks to the Author):

The authors have provided clear and detailed responses to each of my concerns, and they have made appropriate adjustments to the manuscript. These modifications have improved the work's impact and clarity while thoroughly and persuasively addressing my earlier criticisms. The revisions they have made significantly improve the manuscript. I have no more comments on this work, and I believe it is now well-suited for publication in Nature Communications.

We sincerely thank the reviewer for the thorough evaluation of our revised manuscript and for the constructive feedback throughout the review process.

We are grateful for the positive assessment and are pleased that the revisions have improved the clarity, and impact of the work. We appreciate the reviewer's thoughtful engagement, which has helped strengthen the manuscript.

Reviewer #4 (Remarks on code availability):

The code provide a README file. And i could install the dependencies of this code. However, due to the absence of the original files required to run the code, it is not possible to confirm whether the script can be reproduced.

We thank the reviewer for carefully assessing the code availability.

We would like to clarify that the input files are not hosted on GitHub because they are publicly available through established repositories. The README has now been updated to

include direct links and accession numbers to the original datasets required to run the analysis.

In addition, the scripts rely primarily on established and publicly available packages, and the specific package versions used are now clearly listed to ensure reproducibility.